# Data Shifts Hurt Chain-of-Thought: A Case Study on Parity Learning

**Lang Yin** *langyin2@illinois.edu*
*University of Illinois Urbana-Champaign*

**Debangshu Banerjee** *db21@illinois.edu*
*University of Illinois Urbana-Champaign*

**Gagandeep Singh** *ggnds@illinois.edu*
*University of Illinois Urbana-Champaign*

**Reviewed on OpenReview:** *https://openreview.net/forum?id=YWFmIoHP5y*

## Abstract

Chain-of-thought (CoT) has been widely used in large language models and proven to be effective in improving the quality of outputs. Meanwhile, recent theoretical works study a narrower mechanism in which transformers autoregressively generate supervised intermediate computation tokens. It has been rigorously confirmed that under this formal mechanism, structured CoT decompositions enable simple transformers to solve the uniform $k$-parity problem, a problem that cannot be efficiently solved by any direct input-output learning algorithm based on gradient updates. However, existing analysis of this parity construction assumes uniformly generated training inputs and correct signals at every intermediate step. In this work, we study how this specific construction behaves when these assumptions are relaxed. Focusing on a particular binary-tree decomposition for $k$-parity, we investigate the joint effects of imbalance on input distribution and adversarial sign flips of intermediate labels with a simple transformer architecture, teacher forcing and one gradient update. We derive a necessary and sufficient condition for successful learning that depends jointly on the distribution imbalance parameter and the structure of the flipped labels. Surprisingly, our main result implies that input distribution imbalance becomes harmful in our case with the help of CoT, although such imbalance was helpful for the same task under the direct input-output regime. These results characterize the robustness of this parity construction and do not directly establish general conclusions about natural-language CoT prompting or large language models. Code is available at https://github.com/yinlang95/Data-Shift-Impact-on-CoT-for-k-parity.

## 1 Introduction

Large language models (LLMs) based on the transformer architecture has achieved tremendous success in the area of artificial intelligence (Vaswani et al., 2017). However, without intermediate guidance or supervision, they do not perform well especially on complex reasoning problems which require rigorous logical steps Sakarvadia et al. (2023). Chain of Thought (CoT) has empowered LLMs to a large extent (Wei et al., 2022), making them much more capable at multi-step reasoning (Nye et al., 2021; Wei et al., 2022; Zelikman et al., 2022; Lightman et al., 2024), and more effective against hallucinations (Dhuliawala et al., 2024). From a theoretical point of view, CoT has recently been proven to fundamentally improve the power of transformers from a complexity-theoretic perspective (Merrill and Sabharwal, 2024; 2023; Merrill et al., 2022; Li et al., 2024b). Several works have applied the CoT mechanism to solve concrete mathematical problems that are hard for primitive models, such as function classes via in-context learning (Li et al., 2023; Bhattamishra et al., 2024) and the $k$-parity problem (Kim and Suzuki, 2025).

However, existing theoretical analyses of this parity construction assume uniformly generated inputs and correct supervision at every intermediate computation step. In this work, we investigate how this specific binary-tree decomposition mechanism with teacher forcing behaves when the training input distribution is imbalanced and when selected intermediate labels are corrupted.

In this paper, we study a narrower formal notion of chain-of-thought (CoT) used in the theoretical literature, a model that autoregressively generates a prescribed sequence of intermediate computational tokens, and these tokens are supervised during training via teacher forcing (Wies et al., 2023; Zhang et al., 2024; Huang et al., 2024; Kim and Suzuki, 2024; 2025). Thus, throughout this work, unless specified otherwise, "CoT" refers to this specific, formal intermediate-token computation mechanism for the $k$-parity problem, rather than to natural-language prompting, pretraining, or semantic reasoning in modern large language models. Exact details of this formal mechanism will be thoroughly introduced in Section 3.

**Motivations.** In this paper, we focus on answering the following major question:

*Is CoT still effective under data shifts for solving k-parity using a provably successful mechanism?*

To tackle this question, we choose the "**generalized**" $k$-parity problem as a platform, where the term "**generalized**" refers to a broad class of input generating distributions. They will be rigorously defined in Section 2, and they roughly divide the classes of $k$-parity problems into an "*easy*" class and "*hard*" class; the level of "*easiness*" (or "*hardness*") can be precisely, quantitatively characterized by the parameter of input generating distributions. At a high level, a non-uniform distribution makes the problem easier by "leaking" information on the locations of target bits. Another type of shift is data poisoning in the training data of CoT steps. In this work, we investigate the joint impact of both training distributions and data poisoning in those steps on the performance of CoT for the generalized k-parity problem. The answers revealing this three-way relationship is our main contribution in this work. The major question can further be divided into the following more specific ones, and they are the key topics being investigated in our work:

1. *Does more information leak help the algorithm to identify the correct positions of relevant bits?*
2. *How severe is the impact of data poisoning? We divide this question into two more specific ones:*
    (a) *What is the threshold on the level of poisoning that learning can tolerate?*
    (b) *Is there a specific pattern of corruption that harms learning?*
3. *How do both types of shifts affect the training if they concurrently exist?*
4. *Can we explain the mechanism of such effect?*

All five questions will be answered in Section 4.2.

**Choice of the $k$-parity problem.** The $k$-parity problem aims to guess the sign of the product of $k$ selected bits from $\{-1, 1\}$ among a large number of bits, and consequently identify the relevant positions involved in the product. We select it for two main reasons.

1. The major theme of our paper is about the impact of distribution shifts and data poisoning. Any shift on binary inputs for the $k$-parity problem can be precisely quantified. At every step and every position, an entry has only one correct value given a number of bits, so anything other than the true value is poisoning.

2. The quality of a parity predictor can be objectively assessed as the correctness of the output has an absolute criterion.

**Contributions.** Our contributions are summarized as follows.

1. Theorem 4.1 4.1 shows that the imbalanced $k$-parity problem is "easy": It indeed can be efficiently solved by a one-layer transformer in one gradient update without CoT, and the optimization landscape is benign.
2. Next, we reveal the joint impact of the distribution shift and data poisoning on the performance of the predictor trained by the successful CoT decomposition of the $k$-parity problem introduced in (Kim and Suzuki, 2025). This three-way relationship is compressed into one statement in Theorem 4.2, and characterizes a necessary and sufficient condition on the amount of distribution shift and data poisoning to ensure successful training. This result has several implications.

- The tolerance of corrupted CoT training samples is only $O(1/k)$, making the task decomposition vulnerable against data poisoning.
- Surprisingly, distribution shift always hurts under this mechanism: A higher degree of shift always leads to worse training performance. In our setting, recall that non-uniform distributions leak information on the location of target bits, so intuitively it should help the predictor to learn. However, our result suggests the opposite, and Corollary 4.4 eliminates the possibility of successful learning under this task decomposition mechanism when the locations are exposed to the maximum extent.

## 2   Related works

**Empowerment of CoT.**   Starting from 2022, a line of work investigates the expressive powers of transformers from a complexity theoretic perspective. The most recent and comprehensive works include (Merrill and Sabharwal, 2024) and (Li et al., 2024b). The first paper provides a comprehensive complexity-theoretic relationship between CoT steps and computational power. Almost concurrently, the second paper proves tighter upper bounds for constant-precision transformers. Together, these two works confirm that, with sufficient (polynomial) CoT steps, transformers break their original upper bound in computation and can compute any problem in `P/poly`. Beyond theoretical soundness of those works, more concrete implementations on CoT are also being studied, including concrete training paradigms (Li et al., 2024a) and interactions with inference-time search and reinforcement learning fine-tuning (Kim et al., 2025).

**Empirical discoveries of limitations on CoT.**   Despite both theoretical and empirical successes, recent empirical works also revealed that sometimes CoT may worsen the performance (Shaikh et al., 2023; Kambhampati et al., 2024). The CoT mechanism has been shown to improve performance mainly on mathematical and logical tasks, but less so for other tasks (Sprague et al., 2025). For tasks where thinking can make human performance worse, the harm caused by overthinking also holds for models with CoT (Liu et al., 2024). It was also found recently that transformers can still solve problems with meaningless filler CoT tokens (Pfau et al., 2024).

Recent studies further suggested that the effects of CoT depend on the structure of the reasoning trajectory and the type of perturbations. Lu et al. (2025) analyzed the dual role of CoT in jailbreak robustness and showed that reasoning alone may not be enough to prevent harmful behaviors. Feng et al. (2025) found that, longer reasoning traces or increasing reviewing alone may not be always beneficial; instead, identifying failed reasoning branches might more reliably indicate incorrectness. In code-generation tasks, (Liu et al., 2026) showed that CoT does not provide uniform robustness gains, instead structurally sensitive points at which input perturbations may destabilize the reasoning trajectory. These studies focus on natural-language CoT in modern reasoning models, while our work studies a prescribed, teacher-forced intermediate-token computation for the $k$-parity problem. We therefore view them as empirical motivation for studying CoT fragility, rather than direct validation of our theoretical results.

**Label noise and poisoning of reasoning supervision.**   Most past works on learning with noisy labels focus settings where supervised targets are stochastic corruptions of underlying ground-truth labels, and develop methods such as noise-transition modeling and loss correction for robust learning (Patrini et al., 2017; Song et al., 2020). However, data poisoning has an adversarial nature, and therefore allows both stochastic (like previous studies) and strategic modifications of training examples in order to corrupt the behavior of the learned model (Steinhardt et al., 2017).

Our corruption model is relevant to both settings but is not standard classification label noise. Instead of assigning only one noisy output label to each training example, our setting allows sign flips at specified, perhaps strategically selected nodes on a parity computation tree. More details on our characterizations of such corruptions will be discussed in Section 4.

Some recent works studied poisoning attacks that specifically target on reasoning traces. Foerster et al. (2026) introduced decomposed reasoning poisoning, where an attacker modifies the reasoning path while leaving the prompt and final answer clean. They found that CoT introduces additional attack surfaces that make the system more vulnerable, although the separation between intermediate reasoning and final-answer

generation may also make the attack more difficult. Chaudhari et al. (2026) introduced Thought-Transfer, an indirect clean-label attack that modifies reasoning traces, preserves the original queries and correct final answers and consequently aims to induce targeted behavior on other tasks. These papers study poisoning of natural-language reasoning traces in large models, on the other hand we provide here an exact theoretical analysis of binary process-label corruption in a fixed task decomposition. The two lines of work do not conflict but concern different models and threat settings.

**Parity learning and task decomposition.** It has been shown that if, with a positive probability, the relevant bits are uniformly 1 or $-1$, then this "**imbalanced**" $k$-parity problem can be solved by an one-layer neural network Daniely and Malach (2020). However, if all inputs are uniformly generated, then this uniform $k$-parity problem has been proven not to be solvable by any input-output learning algorithm based on gradient updates (Shalev-Shwartz et al., 2017; Shamir, 2018). Recently, thanks to developments of CoT, several works have made significant progresses on solving uniform $k$-parity with task decompositions as CoT steps. Success has been achieved for recurrent neural networks in Wies et al. (2023), and they designed a task decomposition of $k$-parity into $k-1$ structured steps. Afterwards, Kim and Suzuki (2025) extended their results to autoregressive transformers.

**Relation to Kim and Suzuki (2025).** Our work is a major extension of (Kim and Suzuki, 2025) as our work is based on their teacher-forced, one-gradient-update result for learning parity from clean and uniformly generated data. We adopt their one-layer, single-head softmax architecture, absolute positional encodings, the specific binary-tree decomposition into 2-parity computations, teacher-forcing objective, and quantitative analysis on gradient updates assigned to children and non-children of each intermediate node.

Nevertheless, there are two principal differences between our work and theirs. First, Kim and Suzuki analyze the teacher-forced construction under the uniform input distribution, while we study the generalized distribution so the data may not be uniformly generated. Without uniformity, multilinear interactions with zero mean under the uniform case now have nonzero means. These additional quantities complexify the child-versus-non-child gradient comparisons and consequently demand a more dedicated moment analysis. Second, Kim and Suzuki assumed correct, uncorrupted intermediate supervision in this result, but we allow adversarial sign flips at arbitrary steps of the computation. We provide a formal measurement of the poisoning by introducing rigorous metrics, and show that two poisonings with an identical amount may lead to completely different consequences, depending on the poisoning structure, which can be measured by our metrics. Combining these two extensions leads to our joint necessary-and-sufficient condition for successful learning under input-distribution imbalance and corrupted process supervision.

In conclusion, their work was based on assumptions of uniformity and correctness (of data), and we extend by breaking the two assumptions by introducing non-uniform data generations and adversarial data poisoning. However, all our theoretical and empirical conclusions in this work only apply for this simple architecture and the chosen task decomposition. In contrast, the separate no-CoT hardness result (Theorem 2) in (Kim and Suzuki, 2025) is formulated for the broad class of algorithms with access to approximate gradient oracles.

## 3 Problem setup

**Notation.** We write $[n] := \{1, \cdots, n\}$ for any positive integer $n$. The multi-linear inner product of vectors $\boldsymbol{z}_1, \ldots, \boldsymbol{z}_r \in \mathbb{R}^n$ for any $r \in \mathbb{N}$ is denoted as $\langle \boldsymbol{z}_1, \cdots, \boldsymbol{z}_r \rangle := \sum_{i=1}^n z_{1,i} \cdots z_{r,i}$. In particular, $\langle \boldsymbol{z} \rangle = \boldsymbol{z}^\top \mathbf{1}_n$ and $\langle \boldsymbol{z}_1, \boldsymbol{z}_2 \rangle = \boldsymbol{z}_1^\top \boldsymbol{z}_2$. The transformer will be denoted by a function $\mathrm{TF}(\cdot)$. Unless specified, each binary vector $\boldsymbol{x}$ represents the ground truth, and $\hat{\boldsymbol{x}}$ is the generated vector by the transformer. The definition of data poisoning or data corruption will be formally presented later, but if $\boldsymbol{x}$ is injected with poisoning, then it is denoted by $\tilde{\boldsymbol{x}}$.

### 3.1 The parity problem

Let $d \geq k \geq 2$ be integers, and $P$ be an arbitrary subset of $[d]$ with $k$ elements. In this paper, we study the $k$-parity problem, where the output of the target parity function is $y = \prod_{j \in P} x_j$, so the function value called *parity*, depending on the coordinates at the locations determined by $P$. Here, for an input vector

$x = (x_1, \ldots, x_d) \in \{-1, 1\}^d$, $x_j$ denotes the $j$-th coordinate of $x$. Given $n$ samples $(\boldsymbol{x}^i, y^i)_{i \in [n]}$, our goal is to construct a predictor that outputs the parity of any test input from $\{\pm 1\}^d$. We assume $k = \Theta(d)$.

It is known that, if all inputs $\boldsymbol{x}^i$ of dimension $d$ are uniformly generated, then this "**uniform**" $k$-parity problem is fundamentally difficult and cannot be solved in polynomial time by any finite-precision gradient-based algorithms (Wies et al., 2023). Recently, it was proven that the uniform $k$-parity problem can be solved by transformers with $\log k$ reasoning steps (Kim and Suzuki, 2025).

On the other hand, for a particular kind of imbalanced distribution on the input bits for training, the $k$-parity problem is proven to be solvable by neural networks (Daniely and Malach, 2020). The "imbalanced" distribution is defined as the following: For any number $\rho \in [0, 1]$ and a subset $P$ of $[d]$, the distribution $\mathcal{D}_\rho^P$ is a distribution on the input bits such that

- With probability $\rho$, all $d$ bits are uniformly generated.

- With probability $1 - \rho$, all bits in $[d] \setminus P$ are uniformly generated, but the bits in $P$ are all 1 with probability $1/2$, and all $-1$ also with probability $1/2$.

If $\rho = 1$, $D_\rho^P = D_1^P$ reduces to the uniform distribution. Intuitively, any distribution $D_\rho^P$ with $\rho < 1$ leaks information for the relevant bits and consequently makes the parity problem easier. As we will show later in Section 4.1, this imbalanced $k$-parity can also be solved by a one-layer transformer. If the value of $\rho$ is not specified, we categorize such a problem a "**generalized**" $k$-parity problem.

## 3.2 Chain of Thought (CoT)

In this paper, we use the term chain-of-thought (CoT) following the formal theoretical literature on transformers (Wies et al., 2023; Kim and Suzuki, 2025). Specifically, CoT refers to autoregressive generation of intermediate tokens that decompose a complex computation into a sequence of simpler computational tasks.

This notion is not identical with CoT in natural-language CoT prompting in large language models (Wei et al., 2022). Our analysis focuses on the formal intermediate-token computation mechanism rather than language reasoning, pretraining, or prompting.

**Task decomposition as CoT.** As in (Wies et al., 2023) and (Kim and Suzuki, 2025), we assume $k = 2^v$ for simplicity, where $v \in \mathbb{N}$. The assumption $k = 2^v$ is for analytical convenience. For a general $k$, we can pad the computation tree with dummy leaves whose values are fixed to 1 until reaching the next power of two. Since multiplying by 1 does not change the parity value, this padding does not change the target function. We therefore focus on the $k = 2^v$ case without loss of generality.

The CoT protocol decomposes the $k$-parity problem as a sequence of 2-parity problems. Visually, this is expressed as a complete binary tree of height $v$ and $2k - 1$ nodes, shown in Figure 1. The lowest level on this tree contains $k$ nodes from $P$, and the remaining $d - k$ irrelevant bits are isolated vertices and not part of the tree. All remaining nodes represent reasoning steps, and are labeled as $x_{d+1}, \ldots, x_{d+k-1}$. The next level above contains $k/2$ nodes, then $k/4$, and so on until at level $v$, the unique node $x_{d+k-1}$ is the final prediction of the parity value. For each $d < m < d + k - 1$, $x_m$ must have exactly two children and they are denoted by $c_1[m]$ and $c_2[m]$. At the same time, it must have exactly one parent and it is denoted by $p[m]$. The height of the tree is denoted as $h[m]$, the length of the longest path in the graph. All $d$ nodes corresponding to $d$ inputs are located at level zero; the height of any other node is difference between $h[m]$ and the number of edges from itself to the root.

**Transformer parameters.** We study an one-layer transformer architecture with absolute positional encoding and a single-head softmax attention layer, followed by a feedforward layer explained in the next paragraph. During the encoding process, in addition to $d$ input tokens, additional $k - 1$ dummy tokens of zero vectors are added, and each token $\boldsymbol{x}_j$ for $j \in [d + k - 1]$ is concatenated with the one-hot positional encoding $\boldsymbol{e}_j \in \mathbb{R}^{d+k-1}$ to form the final input $\boldsymbol{p}_j = (\boldsymbol{x}_j^\top \boldsymbol{e}_j^\top)^\top \in \mathbb{R}^{n+d+k-1}$ to the attention layer. The first $n$ columns of the key matrix $\mathbf{K}$ and query matrix $\mathbf{Q}$ are set to zero so the attention scores only depend on

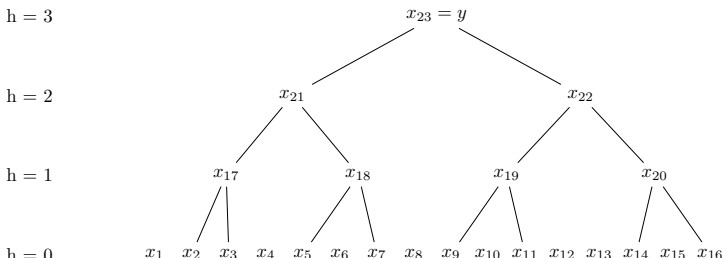

Figure 1: A hierarchical decomposition of an 8-parity problem for $d = 16$. Over here, $x_{17} = x_2 x_3$, so $c_1[17] = 2$, $c_2[17] = 3$, $p[17] = 21$, and $h[17] = 1$.

the positional encodings. The following formula describes the reparameterization of $\mathbf{K}$ and $\mathbf{Q}$ by another $(d + k - 1) \times (d + k - 1)$ matrix $\mathbf{W}$, and the construction of value matrix $\mathbf{V}$.

$$\mathbf{K}^\top \mathbf{Q} = \begin{bmatrix} \mathbf{0}_{n \times n} & \mathbf{0}_{n \times (d+k-1)} \\ \mathbf{0}_{(d+k-1) \times n} & \mathbf{W} \end{bmatrix}, \quad \mathbf{V} = \begin{bmatrix} \mathbf{I}_{n \times n} & \mathbf{0}_{n \times (d+k-1)} \end{bmatrix}.$$

This reparameterization is common in previous literature to make analysis tractable and emphasize positional encoding (Zhang et al., 2024; Huang et al., 2024; Kim and Suzuki, 2024; 2025).

**Feedforward layer.** The feedforward layer carries a fixed link function $\phi : [-1, 1] \to [-1, 1]$, applied element-wise. To exploit the decomposition of our task into 2-parities, we choose $\phi$ such that $\phi(0) = -1$, $\phi(\pm 1) = 1$ so that sums are converted into parities, i.e. $\phi(\frac{a+b}{2}) = ab$ for $a, b \in \{\pm 1\}$. Moreover, we require symmetry of $\phi$, and that $\phi'(0) = 0$. Specifically, we choose the following function.

$$\phi(x) = \begin{cases} d^3 x^2 + d^{-3} - 1, & x \in (-d^{-3}, d^{-3}); \\ 2|x| - 1, & \text{otherwise.} \end{cases} \tag{1}$$

The choice of this function ensures that $\phi$ is differentiable everywhere on $[-1, 1]$. There is a discrepancy on $\phi(0) = d^{-3} - 1$ instead of $-1$, but the gap will be bounded as perturbations and approach to zero as $d$ becomes large.

**Specific CoT process.** For our case on $k$-parity, all input bits $\boldsymbol{x}_1, \ldots, \boldsymbol{x}_d$ are fixed, and the positions of later steps before generation are null. The first intermediate token, $\hat{\boldsymbol{x}}_{d+1}$, is generated next and staying the same value through the entire remaining process. Similarly, the next token is generated by $\hat{\boldsymbol{x}}_{d+2} = \mathrm{TF}^{(2)}(\boldsymbol{x}_1, \cdots, \boldsymbol{x}_d, \hat{\boldsymbol{x}}_{d+1}; \mathbf{W})$, where $\mathbf{W}$ is the transformer weights. Finally, the final prediction is computed by repeating the computation for $k - 1$ times:

$$\boldsymbol{y} = \mathrm{TF}^{(k-1)}(\boldsymbol{x}_1, \ldots, \boldsymbol{x}_d, \hat{\boldsymbol{x}}_{d+1}, \ldots, \hat{\boldsymbol{x}}_{d+k-2}; \mathbf{W}) \tag{2}$$

**Teacher forcing.** For CoT implementations, we utilize *teacher forcing* in our training process. Teacher forcing is a form of process supervision, where in addition to the final prediction, ground truth labels for CoT steps are provided during training. Thus, the accuracy of each CoT step can be measured. Given $n$ samples and model weights $\mathbf{W}$, the total loss takes every position $d + 1 \leq m \leq d + k - 1$ into account and is defined as

$$L(\mathbf{W}) = \frac{1}{2n} \sum_{m=d+1}^{d+k-1} ||\hat{\boldsymbol{x}}_m - \boldsymbol{x}_m||^2 = \frac{1}{2n} \sum_{m=d+1}^{d+k-1} ||\phi(\hat{\boldsymbol{z}}_m) - \boldsymbol{x}_m||^2, \quad \boldsymbol{z}_m = \sum_{j=1}^{m-1} \sigma_j(\boldsymbol{w}_m) \boldsymbol{x}_j, \tag{3}$$

where $\sigma_j(\boldsymbol{w}_j)$ are softmax attention scores.

**CoT data corruption.** Intuitively, if all CoT steps are correct for training, then indeed the final predictor will accurately reflect the true target function thanks to correct decomposition. However, correctness of CoT

steps relies on correct "ground truth" labels during the training steps. If a high amount of such tokens are false, because of either oversight or malicious attacks, then naturally, one may infer that the quality of those intermediate steps may deteriorate. In our case, the ground truth labels are either 1 or $-1$, and corruption refers to flipping the signs of some inputs.

**Scope of analysis.**  All theoretical results in this paper focus on the specific CoT mechanism defined above: The binary-tree decomposition of $k$-parity, the one-layer, single-head transformer architecture with absolute position encodings, the fixed feedforward function, teacher forcing, all-zero initializations, and one-step gradient update.

Our proofs use these choices explicitly. All-zero initialization makes the initial attention weights uniform, the absolute encodings allow us to compare the gradients assigned to child and non-child nodes, and the fixed feedforward function converts the average of two child values into their 2-parity. The one-step training analysis reduces successful learning to those child-versus-non-child gradient update comparisons.

Therefore, our conclusions are consequences of this formal CoT mechanism rather than general statements aboutnatural-language CoT prompting or large language models. They do not directly establish any architecture-agnostic hardness results or absolute limitations on natural language CoT.

## 4 Theoretical results

### 4.1 Imbalance is easier than uniformity for transformers without CoT

In this section, we assume $\rho < 1$ so information on relevant bits in $P$ are leaked. We assume $\rho$ is never too small nor too large, so $\rho = \Theta(1)$. The goal of this section is to show that the imbalanced problem is indeed solvable by a simple, one-layer transformer with a softmax attention layer. Theorem 4.1 is the specific statement of this result. The proof will be presented in Appendix B.

**Theorem 4.1.** *Given $n$ samples where $n = \Omega(d^{2+\epsilon})$, with probability at least $1 - \exp(-d^{\epsilon/2})$, a learning rate $\eta = \Theta(d^\epsilon)$ and all-zero initializations, the predictor $\hat{y}$ after one-step update satisfy $|\hat{y} - y| \leq O(d^{-1+\epsilon})$ for any given input $\boldsymbol{x} \in \{\pm 1\}^d$ and $y = \prod_{r \in P} x_r$.*

*Proof sketch.* The proof involves explicitly computing the gradient with respect to each weight $w_{j,m}$, and utilizing the large differences on the gradients between relevant and irrelevant bits (whether $j \in P$ or not).

Here is an example of interaction: $\langle \boldsymbol{x}_{d+1}, \hat{\boldsymbol{z}}_m, \hat{\boldsymbol{z}}_m \rangle = \langle \boldsymbol{x}_{d+1}, \boldsymbol{x}_\alpha, \boldsymbol{x}_\beta \rangle / d^2$. If $\alpha, \beta \in P$, then for each data sample, the parity $x_{d+1} x_\alpha x_\beta = \prod_{i \in P \setminus \{\alpha,\beta\}} x_i$ is a random variable with expectation $1 - \rho > 0$ if $\rho < 1$. If all bits are uniformly generated, i.e. $\rho = 1$, then this variable has mean zero. Since the training distribution $\mathcal{D}_\rho^P$ is imbalanced, the variable has a positive mean. But if at least one of $\alpha$ and $\beta$ is not in $P$, such variables are bounded with in a small value. With a sufficiently large $d$ and $n$, the computation leads to positive weight updates for relevant bits, and negative updates for others. Such a huge gap allows an attention layer to identify relevant bits and make correct predictions. $\qquad\square$

### 4.2 Applying CoT for the generalized problem

In this section, the value of $\rho \in [0,1]$ is not restricted, and we focus on the impact of CoT decomposition on solving the generalized $k$-parity problem under two types of nuances: (1) the information leaked (regulated by $\rho$), and (2) the level of data poisoning, defined in Section 3.

Our main result Theorem 4.2 characterizes an equivalent condition on successful training with this CoT decomposition with respect to (1) the distribution shift $\rho$ and (2) the quantity and structure of data poisoning. Before stating the theorem, we first define a few ingredients for the characterization.

**Set of poisoning.**  We first formally quantify the data poisoning on different positions and their interactions. Given $n$ data samples of dimension $d + k - 1$ including the CoT steps, for each node $i \in \{d+1, \ldots, d+k-1\}$, let $U_i \subseteq [n]$ be the set of indices with corrupted samples. For any two nodes $a, b \in \{d+1, \ldots, d+k-1\}$,

define the set $I_{a,b} = U_a \cap U_b$. For any $i \in I_{a,b}$, coordinates $a$ and $b$ in the training sample $\boldsymbol{x}^i$ are both flipped by multiplying $-1$, so the effect of poisoning cancels out. For three nodes $a, b, c \in \{d+1, \ldots, d+k-1\}$, we define the following set:

$$U_{a,b,c} = \{(U_a \cup U_b \cup U_c) \setminus (I_{a,b} \cup I_{a,c} \cup I_{b,c})\} \cup (U_a \cap U_b \cap U_c).$$

For two nodes $a$ and $b$, the set $U_{a,b} = (U_a \cup U_b) \setminus (U_a \cap U_b)$ is defined with the same rationale. See Figure 2 and 3 for visualizations. We will denote $q_{a,b,c} = \frac{|U_{a,b,c}|}{n}$ for the quantity of poisoning for this triple.

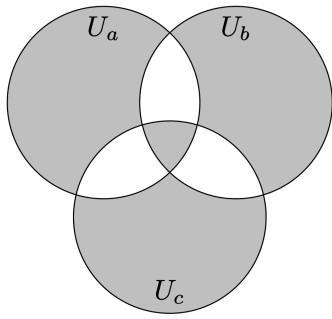
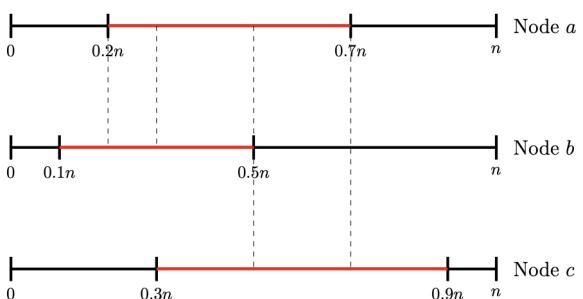

Figure 2: Given three sets $U_a$, $U_b$, and $U_c$, the set $U_{a,b,c}$ is the union of all three sets excluding elements that belong to exactly one intersection of two sets. The shaded region represents the impactful corruption. Corruption in the white region is canceled out.

Figure 3: Each axis represents samples of nodes $a$, $b$, and $c$. Red segments show flipped samples. Flips from $0.2n$–$0.3n$ and $0.5n$–$0.7n$ cancel out due to two flips. Other red segments are harmful: flipped at one node ($0.1n$–$0.2n$, $0.7n$–$0.9n$) or all three nodes ($0.3n$–$0.5n$).

**Impact of poisoning and distribution shift for parity learning.**  Exact quantities of gradient updates $\partial L / \partial w_{j,m}$ for every $1 \le j < m$ depend on the indices of $j$ and $m$. Those quantities are necessary to characterize the poisoning and state our main result Theorem 4.2. In particular, we need to compute $\partial L / \partial w_{j',m}$ where $j'$ is one of the two children of $m$, and $\partial L / \partial w_{j'',m}$ where $j'' < m$ but not a child of $m$. The two updates should have a large difference so the model can recognize the children. As we will discuss in the next paragraphs, poisoning and distribution shift would alter the values of the updates and may consequently mislead the model.

The impact of corruption depends on the interaction among the parent node and its children. In particular, for a parent-children triple $(m, c_1[m], c_2[m])$, corruption at an even number of these nodes cancels out, while corruption at an odd number of nodes changes the sign of this multilinear interaction. Therefore, the effective corruption level is determined not by the total number of corrupted nodes, but by the fraction of samples with an odd number of flips in the relevant interaction. Detailed examples are provided in Appendix A.

The exact expressions for gradient update differences depend on the heights of a thinking step $m$ and the previous step $j$ being compared. There are four cases in total here. To understand them, recall that the CoT mechanism succeeds only if the attention layer assigns substantially larger weights to the two children of every intermediate node than to other previous tokens. Therefore, our analysis compares the gradient update of a correct child of $m$ with that of an incorrect non-child node $j$. The exact expressions depend on whether $m$ and $j$ are input nodes or intermediate computation nodes. We now enumerate the four cases:

1. The height of $m$ is one, and the height of $j$ is zero. In this case, a computation node directly multiplies two input bits, and the competitor is an irrelevant input token.
2. The height of both $m$ and $j$ is one. In this case, both the target node and the competing token are intermediate nodes.
3. The height of $m$ is greater than one, and the height of $j$ is zero. The target node multiplies lower-level computation tokens while the competitor is an input token.

4. The height of $m$ is greater than one, and the height of $j$ is greater than zero. Both the target and competitor are intermediate computation tokens.

**Examples for the cases.** We illustrate each of the four cases using Figure 1. Nodes $x_{17}, x_{18}, x_{19}, x_{20}$ have height one, $x_{21}$ and $x_{22}$ have height two, and the final step $x_{23}$ has height three. If $m = 19$ and $j = 3$, then this is an example of case 1 above. Similarly, $(m, j) = (19, 17)$ corresponds to case 2, $(m, j) = (21, 12)$ is an example of case 3, and finally $(m, j) = (23, 17)$ represents an instance of case 4.

We now spell out the exact expressions for gradient update differences for all four cases above. The expressions for cases 2 and 3 are long so we define two functions to represent them compactly.

1. If $h[m] = 1$ and $h[j] = 0$, the difference between gradient update against the children of $m$ is $-\frac{2\rho(1-2q_m)}{(m-1)^2}$.

2. If $h[m] = h[j] = 1$, the difference is $G_{h[m]=1}(m, j, \rho) + (1 - \rho)S(m, j)$, where

$$G_{h[m]=1}(m, j, \rho) = -\frac{2(1 - 2q_m)}{(m-1)^2} - \frac{2(k-1)(1 - 2q_m)}{(m-1)^2}(1 - \rho) + \sum_{\alpha=d+1}^{m-1} \frac{2(1 - 2q_{m,\alpha,j})}{(m-1)^2}(1 - \rho). \tag{4}$$

and

$$S(m, j) = -\frac{2(k-1)}{(m-1)^2} + \sum_{d<\alpha<m, \alpha\neq j} \frac{2(1 - 2q_\alpha)}{(m-1)^2} + \sum_{\alpha=d+1}^{m-1} \frac{8k(1 - 2q_\alpha)}{(m-1)^3} - \sum_{\alpha,\beta=d+1}^{m-1} \frac{4(1 - 2q_{\alpha,\beta,j})}{(m-1)^3} - \sum_{\alpha,\beta\in P} \frac{4(1 - 2q_m)}{(m-1)^3}. \tag{5}$$

3. If $h[m] > 1$ and $h[j] = 0$, the difference is $G_{h[m]>1}(m, j, \rho) - (1 - \rho)S(m, j)$, where

$$G_{h[m]>1}(m, j, \rho) = -\frac{2(1 - 2q_{m,c_1[m],c_2[m]})}{(m-1)^2} - \sum_{d<\alpha<m, \alpha\neq c'[m]} \frac{2(1 - 2q_{m,c[m],j})}{(m-1)^2}(1 - \rho) + \frac{2k(1 - 2q_m)}{(m-1)^2}(1 - \rho), \tag{6}$$

and $S(m, j)$ is the same function used in the previous case.

4. If $h[m] > 1$ and $h[j] > 0$, the difference is $-\frac{2\rho(1-2q_{m,c_1[m],c_2[m]})}{(m-1)^2}$.

**Interpretation of $B_m$.** For each intermediate node $m$, we introduce a quantity $B_m$ which will play a critical role in our main theorem. The quantity $B_m$ represents the worst case gradient difference between a correct child and an incorrect previous node. It reveals the strongest competitor that imitates a true child in the attention update and consequently confuses the entire computation. A more negative (larger absolute value) $B_m$ implies that the correct children receive a stronger advantage after gradient update and therefore is easier for the attention layer to distinguish. When $B_m$ approaches zero, the gradient signals of correct children and incorrect nodes become indistinguishable, preventing successful learning. In cases 1 and 4, the magnitude of the leading child-versus-non-child separation is $\frac{2\rho(1-2q)}{(m-1)^2}$, where $q = q_m$ for a height-one node and $q = q_{m,c_1[m],c_2[m]}$ for a higher node. Thus, this separation decreases linearly as $\rho$ decreases or as the impactful corruption rate approaches $1/2$, and it vanishes when $\rho = 0$ or $q = 1/2$. Analyses for cases 2 and 3 are similar with more sophisticated algebra being the only additional complexity.

We now state our main theoretical result in this work. We answer Question 3 by providing a comprehensive statement on the impact of both distribution shift $\rho$ and the poisoning structure among the CoT steps on the final performance of the predictor within the selected CoT decomposition. Such impact leads to an equivalent (both necessary and sufficient) condition on success of training.

**Theorem 4.2.** *Let $n = \Omega(d^{2+\epsilon})$ and $-2 - \epsilon/4 < \mu < 0$ for $\epsilon > 0$. Suppose $d$ is sufficiently large. With softmax attention and all-zero initializations on weights, a transformer with the prescribed CoT mechanism can solve the generalized parity problem with an error rate converging to zero as $d \to \infty$ if and only if all the following conditions hold.*

- *For every $m \in \{d+1, \cdots, d+\frac{k}{2}\}$, the following inequality satisfies:*

$$B_m = \max_{d<j<m} \left\{ -\frac{2\rho(1 - 2q_m)}{(m-1)^2}, \quad G_{h[m]=1}(m, j, \rho) + (1 - \rho)S(m, j) \right\} < -O(d^\mu). \tag{7}$$

- *For every $m \in \left\{ d + \frac{k}{2} + 1, \cdots, m - 1 \right\}$, the following inequality satisfies:*

$$B_m = \max_{d < j < m} \left\{ -\frac{2\rho(1 - 2q_{m,c_1[m],c_2[m]})}{(m-1)^2}, \quad G_{h[m]>1}(m, j, \rho) - (1 - \rho)S(m, j) \right\} < -O(d^\mu). \qquad (8)$$

*In particular, if the conditions above hold for every $m$, then let $B = \max_{d < m < d+k-1} B_m$, for every input $\boldsymbol{x} \in \{\pm 1\}^d$, the true prediction $y$ and the prediction $\hat{y}$ by the trained predictor after one step update with learning rate $\eta = \Theta(d^{-\mu})$ satisfy $|\hat{y} - y| \leq O(d^{-B(\mu-2-\epsilon/4)})$ with probability at least $1 - \exp(-d^{\epsilon/2})$. If the conditions fail for at least one CoT step, then $\lim_{d \to \infty} \mathbb{E}[|\hat{y} - y|] = \Omega(1)$.*

**Vulnerability against poisoning.** The theorem provides a rigid equivalence condition for the algorithm to succeed. Even for one intermediate step $m \in \{d+1, \cdots, d+k-1\}$, if $q_{m,c_1[m],c_2[m]} \geq 0.5$, then the condition in Equation (7) or 8 no longer holds, and consequently the training algorithm would not succeed. Recall that for this task, there are $(k-1)n$ values in the training data set for all intermediate steps, but $0.5n$ flips are sufficient to fail the training. Conclusively, this algorithm has a low poisoning tolerance of $\frac{1}{2(k-1)}$, and this threshold approaches to zero as $k$ become large. This analysis answers Questions 2.(a) and 2.(b) in Section 1.

Regarding distributions shift, Theorem 4.2 leads to an immediate corollary, which reveals a seemingly paradoxical conclusion.

**Corollary 4.3** (Maximum leakage of information). *If $\rho = 0$, i.e. all non-relevant bits are still uniformly generated but all relevant bits are either all $-1$ or $1$, then the prescribed CoT training procedure fails for this decomposition.*

*Proof.* If $\rho = 0$, then $B_m = 0$ for any intermediate node $m$. $\qquad \square$

**Explanation of the "paradox".** If $\rho = 0$, the locations of relevant bits are exposed to the maximum extent, so intuitively, the CoT protocol should be able to solve it even more efficiently than the case when $\rho = 1$. However, within this CoT design, this case leads to an immediate, absolute failure. The exact reason will be briefly outlined in the proof sketch and comprehensively presented in the full proof. At a high level, when $\rho = 0$, the gradient update extracts identical information from *correct* nodes (children) and *incorrect* nodes (non-children) during the CoT steps. However, identifying the location of the children is essential to ensure that the final output is indeed the multiplication of the relevant bits. If errors on this step exist, some relevant bits are multiplied multiple times and therefore the output will be different from the truth.

**Ever-present harm of distribution shift.** The impact of $\rho$ is concrete even if $\rho > 0$. Recall that for the uniform case where $\rho = 1$, we have

$$\begin{cases} G_{h[m]=1}(m, 1) = -\frac{2(1-2q_m)}{(m-1)^2} = -\frac{2(1-2q_{m,c_1[m],c_2[m]})}{(m-1)^2}, \\ G_{h[m]>1}(m, 1) = -\frac{2(1-2q_{m,c_1[m],c_2[m]})}{(m-1)^2}. \end{cases} \qquad (9)$$

Thus, $B_m = -2(1 - 2q_{m,c_1[m],c_2[m]})/(m-1)^2$ for any $m$ if $\rho = 1$. Let $m' = \arg\max_m B_m$, then if $\rho < 1$, denote the values computed in Equation (7) and 8 as $\{B'_m\}_{m=d+1}^{d+k-1}$, observe that $B_{m'} > B_m$. Hence, $B' = \max_m B'_m > B'$ and the convergence rate slows down for every $\rho < 1$. Furthermore, clearly a lower $\rho$ leads to a higher $B_m$. So, we can conclude that distribution shift always damages the training if it exists, and low shift is always better than high shift. This answers Question 1 within our prescribed setting.

**Corollary 4.4** (Simple characterization without distribution shift). *If the training and testing distribution are identical, i.e. $\rho = 1$, then the CoT decomposition succeeds if and only if $q_{m,c_1[m],c_2[m]} \leq 0.5 - O(d^\mu)$ for every $m \in \{d+1, \ldots, d+k-1\}$.*

*Proof.* If $\rho = 1$, then clearly $G_{h[m]=1} + (1-\rho)S = G_{h[m]>1} - (1-\rho)S = -\frac{2(1-2q_{m,c_1[m],c_2[m]})}{(m-1)^2}$. $\qquad \square$

We have answered all of Questions 1, 2 and 3 in the introduction. We conclude this section by a proof sketch of Theorem 4.2, which summarizes the technical analysis that answers Question 4. The entire proof will be presented in Appendix C.

*Proof sketch of Theorem 4.2.* For the *if* direction, we directly compute the gradients, and found that the two quantities in Equation (7) and 8 are gradient differences between correct and incorrect nodes. A low enough value of $B_m$ for every $m$ ensures the two correct nodes have larger weights than incorrect nodes, and the gap must be large enough for the attention layer to distinguish correct and incorrect nodes as $d$ becomes large. As a result, the attention scores $\sigma_j(w_m)$ is close to zero if $p[j] \neq m$, and are close to 0.5 otherwise.

For the *only if* direction, we prove the contrapositive: If the conditions do not hold, i.e. $B_m$ is not low enough for an intermediate step $m$. The key part is still analyzing the gradient update, but since the condition fails for $m$, the gradient update for at least one node $j$ such that $p[j] \neq m$ is now equal or higher than updates for children. Thus, at least one CoT step will not learn an accurate predictor. $\qquad\square$

## 5  Experiments

The statement of Theorem 4.2 holds with a large enough $d$ to overcome low-order error terms and perturbations. In our experiments, we implement a more realistic value of dimension ($d = 128$) and learning rates with an extensive period of training time. We train a simple one-layer transformer with absolute encoding, softmax attention layer and the feedforward layer defined in Section 3.2.

Our first key observation is that, the performance under no distribution shift strictly surpasses any other case with distinct poisoning structure and quantity, with the exception of the case where the first and second CoT steps have 40% of ground truth labels flipped. The high loss for this poisoning structure even without distribution shift is justifiable as the level of impact poisoning is high. We also observe that, even without distribution shift, the empirical poisoning tolerance is not as high as 50%, and the location of poisoning matters. Theoretically, as long as the impactful poisoning rate is below 50%, the outcome should be identical with the case without any poisoning if $d$ is large enough. But such an eligible $d$, as we will show in the proof of Theorem 4.2, must be astronomically large and cannot be empirically tested.

The empirical results strengthens our theoretical discoveries on the relationship between the CoT performance and data shifts. Meticulous assessment of the data shifts is essential to ensure the success of the CoT training with decomposition in Section 3.

Our implementation automatically generates synthetic data from $\{\pm 1\}^{128}$ and randomly selects $k$ relevant bits. Once the inputs are generated and relevant bits are selected, the program then constructs a decomposition tree like the illustration in Figure 1. Next, using the inputs, the program computes the ground truth samples for CoT training by multiplying the correct bits element-wise following the decomposition tree structure.

In addition to standard parameters, we also investigate the ratio of uniformly generated inputs and structure/quantity of data poisoning. After the inputs are generated, the program allows us to input the variable `uniform_prob`, which is $\rho$ defined in Section 4, and then $(1 - \texttt{uniform\_prob})n$ samples will have their coordinates at target bits to be changed to either all $-1$ or $1$, both with probability 0.5. We may also easily inject poisoning with any quantity and structure by editing the list `flip_configurations`.

We experimented over 35 cases with seven variants of poisoning quantity and structure and five values of `uniform_prob`. Figure 4 shows the training results after 5000 epochs in every case when $d = 128$ and $k = 64$.

The feedforward layer function for the transformer is the same function (Equation (1)) in Section 3.2.

The testing data are uniformly generated. Although for general tasks, it is natural to expect that the test error should be large if the training and testing distributions are different. However, in this $k$-parity test, if the training is successful, the predictor is expected to identify the positions of relevant bits, regardless the training distribution where it was learned. Therefore, testing the predictor for all values of $\rho \in \{0, 0.25, 0.5, 0.75, 1\}$ is the only fair measure of the predictor's performance.

## 6  Conclusion

In this work, we analyze the robustness of a formal intermediate-token CoT construction for the $k$-parity problem. In particular, we analyzed the robustness of a formal intermediate-token CoT mechanism for

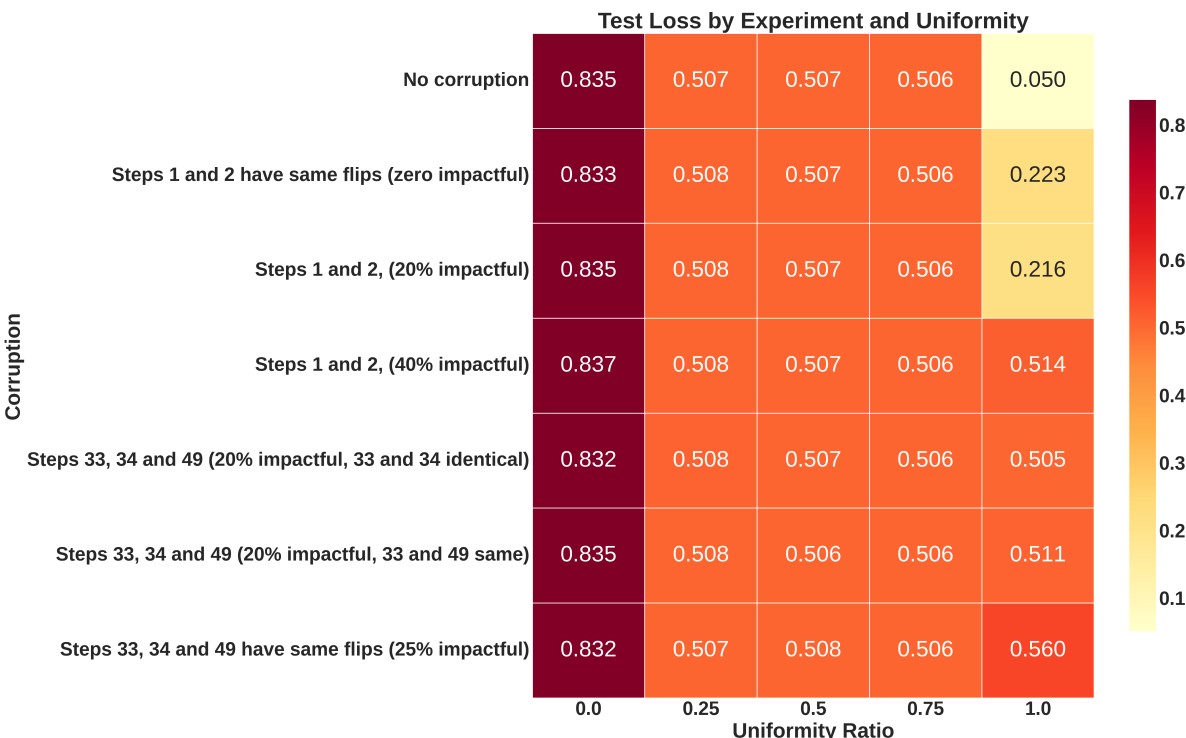

Figure 4: Heatmap of testing loss with respect to both distribution shift and poisoning structure. Each value on the horizontal axis is the ratio of uniformly generated inputs, the same as $\rho$ in Section 4. The number inside each grid is the test loss under that particular circumstance.

the $k$-parity problem with two types of turbulence: imbalance on input distribution and corruption of intermediate computation. Under the mechanism, characterized with the well-defined problem based on binary inputs and outputs, prescribed binary-tree decomposition, one-layer transformer architecture, and one-step training procedure, we derived a **necessary and sufficient** condition for successful learning in terms of the distribution imbalance parameter $\rho$ and the structured corruption rates. We also found that, counterintuitively, the distribution imbalance that was helpful for parity learning weakens the performance of the same learning task with the help of CoT.

**Limitations.** All results in this work are specific to the learning task, decomposition, architecture, and optimization regime analyzed in this paper. The parity problem captures autoregressive generation of intermediate tokens, causal dependence of later computation steps on earlier ones, and teacher-forced process supervision in an exactly analyzable setting. However, it does not capture natural-language reasoning, semantic abstraction, pretraining and prompting, multiple valid reasoning paths, redundancy, or self-correction. Our results do not directly establish conclusions about natural-language CoT or large language models, and possible connections to those broader settings should be viewed only as hypotheses for future study. Within the scope of our setting, our conclusions still depend on the prescribed binary-tree decomposition first introduced by (Kim and Suzuki, 2025), illustrated by Figure 1. Alternative decompositions, such as flatter trees or decompositions with redundant steps, may interact with the same data shifts differently and therefore lead to different conclusions. Extending the analysis to more sophisticated tasks, decompositions, architectures, and optimization procedures is an important potential direction for future work.

**Reproducibility statement.** All assumptions, notations, and technical setup of the theoretical results are included in the main body of the paper, particularly in Sections 3 and 4, and proofs for all claims are included in the Appendices B and C. The code link is provided in the abstract. Figures and implementation details for the experiments are discussed in Section 5 and Appendix D. Our experiments used solely synthetic data.

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

# Technical Appendices and Supplementary Material

## A    Examples of Poisoning

Over here, we provide a few examples of poisoning and the consequences. Recall that the computation of the gradient updates involves the multi-linear interactions $\langle \boldsymbol{x}_m, \boldsymbol{x}_a, \boldsymbol{x}_b \rangle$ for steps $m \in \{d+1, \ldots, d+k-1\}$ and earlier steps $a, b < m$. Observe that, if all three values are correct, we must have $\langle \boldsymbol{x}_m, \boldsymbol{x}_a, \boldsymbol{x}_b \rangle = n$ as the vectors are $n$-dimensional. We illustrate a few examples of concrete poisoning using Figure 1. Consider the triple $(17, 2, 3)$, where 17 is the parent of 2 and 3. Note that the values at positions 2 and 3 are inputs, so they cannot be flipped; therefore, any flip at position 17 (i.e. $\hat{x}_{17} = -x_2 x_3$ instead of the correct value $x_2 x_3$) is harmful to the learning process because the multi-linear interactions lead to $\langle \hat{\boldsymbol{x}}_{17}, \boldsymbol{x}_2, \boldsymbol{x}_3 \rangle = -n$, an incorrect value. On the other hand, the analysis is different for parents at a higher height. For instance, consider the triple $(22, 19, 20)$ and suppose there are 100 data samples. We discuss the impact of three concrete cases of poisoning below.

1. First, we look at the case when an even number of nodes have identical corresponding samples flipped. Suppose nodes 22 and 20 have the samples indexed with 11-20 flipped, i.e. for any $i \in \{11, 12, \ldots, 20\}$, $\hat{\boldsymbol{x}}_{22}^i = -\boldsymbol{x}_{22}^i$ and $\hat{\boldsymbol{x}}_{20}^i = -\boldsymbol{x}_{20}^i$ where values with hat denote the actual values used in computation, and values without hat denote the ground truth values. In this case, the gradient update is not impacted at all: Indeed, for any $i \in \{11, 12, \ldots, 20\}$, the multi-linear product $\langle \hat{\boldsymbol{x}}_{22}^i, \hat{\boldsymbol{x}}_{19}^i, \hat{\boldsymbol{x}}_{20}^i \rangle = \langle -\boldsymbol{x}_{22}^i, \boldsymbol{x}_{19}^i, -\boldsymbol{x}_{20}^i \rangle = \langle \boldsymbol{x}_{22}^i, \boldsymbol{x}_{19}^i, \boldsymbol{x}_{20}^i \rangle = 1$.
2. Now suppose all three nodes have an identical sample $i$ flipped for $i \in \{11, \ldots, 20\}$, then we have $\langle \hat{\boldsymbol{x}}_{22}^i, \hat{\boldsymbol{x}}_{19}^i, \hat{\boldsymbol{x}}_{20}^i \rangle = \langle -\boldsymbol{x}_{22}^i, -\boldsymbol{x}_{19}^i, -\boldsymbol{x}_{20}^i \rangle = -\langle \boldsymbol{x}_{22}^i, \boldsymbol{x}_{19}^i, \boldsymbol{x}_{20}^i \rangle = -1$. Therefore, this sample $i$ would mislead the model as the multi-linear product returns an incorrect value.
3. For the last example, we consider a general case: any node may have an arbitrary collection of samples to be flipped, though those samples may overlap; the exact impact of poisoning needs to be carefully analyzed in such cases. Suppose the position 22 has samples indexed with 11-20 flipped, position 19 has samples 16-25 flipped, and position 20 has samples 91-100 flipped. For any sample $i \in \{16, \ldots, 20\}$, we have $\langle \hat{\boldsymbol{x}}_{22}^i, \hat{\boldsymbol{x}}_{19}^i, \hat{\boldsymbol{x}}_{20}^i \rangle = \langle -\boldsymbol{x}_{22}^i, -\boldsymbol{x}_{19}^i, \boldsymbol{x}_{20}^i \rangle = 1$. So the poisoning of sample $i$ causes no harm. Also, for sample $i' \in \{11, \ldots, 15\}$, we have $\langle \hat{\boldsymbol{x}}_{22}^i, \hat{\boldsymbol{x}}_{19}^i, \hat{\boldsymbol{x}}_{20}^i \rangle = \langle -\boldsymbol{x}_{22}^i, \boldsymbol{x}_{19}^i, \boldsymbol{x}_{20}^i \rangle = -1$. So poisoning here is harmful. The same holds for $i' \in \{91, \ldots, 100\}$. Therefore, although there are thirty flipped tokens in total, only fifteen of them are in fact impactful.

For any parent-children triple $m, a, b$, we denote $q_{m,a,b}$ as the ratio of harmful samples among all samples. If $h[m] = 1$, then we just write $q_m$ because the children of $m$ cannot be poisoned. The values of $q_{22,19,20}$ in our three examples above are 0, 0.1, and 0.15, respectively.

## B    Proof of Theorem 4.1

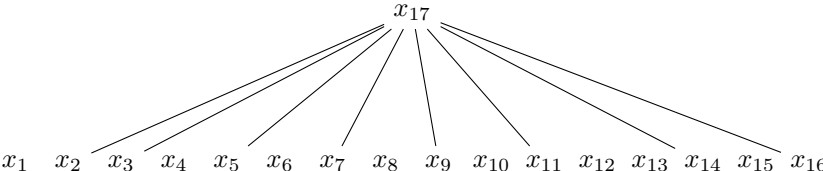

Figure 5: The same parity function where $P = \{2, 3, 5, 7, 9, 11, 14, 16\}$ as one represented by Figure 1, but without CoT decomposition.

By proving Theorem 4.1, we show that the $k$-parity problem can be solved by simple transformers if the data samples are generated via the distribution $D_\rho^P$ for $\rho < 1$.

First, we visually understand the problem. We illustrated the case with CoT using Figure 1. If we omit CoT and use the simple input-output model, then the tree only has depth one, where the root is the final output

and its $k$ children are relevant bits, as illustrated in . Given $n$ samples, the loss function is

$$L(\mathbf{W}) = \frac{1}{2n} \|\hat{\boldsymbol{x}}_{d+1} - \boldsymbol{x}_{d+1}\|_\infty^2. \tag{10}$$

Unlike the loss function for the case with CoT, the function here is a single summand because the only prediction during the process is the output.

Using the softmax attention, some quantities in the original paper preserve. We will need the following expressions. For any $1 \le \alpha, j < d + 1$, denote the $\delta_{j\alpha}$ as the 0-1 indicator on $j = \alpha$, we have

$$\frac{\partial \sigma_\alpha(\boldsymbol{w}_{d+1})}{\partial w_{j,d+1}} = (\delta_{j\alpha} - \sigma_\alpha(\boldsymbol{w}_{d+1}))\sigma_j(\boldsymbol{w}_{d+1}) = (\delta_{j\alpha} - \sigma_j(\boldsymbol{w}_{d+1}))\sigma_\alpha(\boldsymbol{w}_{d+1}); \tag{11}$$

and

$$\frac{\partial \hat{\boldsymbol{z}}_{d+1}}{\partial w_{j,d+1}} = \sum_{\alpha=1}^{d} (\delta_{j\alpha} - \sigma_j(\boldsymbol{w}_{d+1}))\sigma_\alpha(\boldsymbol{w}_{d+1})\boldsymbol{x}_\alpha = \sigma_j(\boldsymbol{w}_{d+1})(\boldsymbol{x}_j - \hat{\boldsymbol{z}}_{d+1}). \tag{12}$$

**Feedforward layer.** Because of the nice properties of the 2-parity problem, we could apply a simple feedforward activation $\phi$ as long as $\phi(0) = -1$ and $\phi(\pm 1) = 1$. However, in this "flat" problem, we must choose a feedforward function $\phi$ to satisfy: $\phi\left(\frac{x_1 + \cdots + x_k}{k}\right) = x_1 x_2 \cdots x_k$. This leads to a few other quantitative requirements:

- $\phi(0) = 1$. Because if $x_1 + \cdots + x_k = 0$, then we have equally many 1's and $-1$'s, so $x_1 \cdots x_k = 1$.
- $\phi\left(\pm \frac{2}{k}\right) = \phi\left(\pm \frac{6}{k}\right) = \cdots = \phi\left(\pm \frac{k-2}{k}\right) = -1$. This is the case when the difference between the numbers of 1's and $-1$'s is odd, and their product is $-1$.
- $\phi\left(\pm \frac{4}{k}\right) = \phi\left(\pm \frac{8}{k}\right) = \cdots = \phi\left(\pm \frac{k}{k}\right) = 1$. This is the case when the difference between the numbers of 1's and $-1$'s is even, and their product is 1.

A plausible choice is $\phi(x) = \cos(0.5k\pi x)$; it satisfies all the properties above. When $x$ is small, using Taylor expansion we can approximate its derivative as $\phi'(x) \approx -0.25(k^2\pi^2)x$, and the remaining terms can be approximated as $O(x^3)$. To simplify the notation, we will write $\phi'(x) = -2cx = -2ak^2x$ where $c = ak^2$ and $a = \pi^2/8$.

Therefore,

$$\frac{\partial L}{\partial w_{j,d+1}}(\mathbf{W}) = \frac{1}{n}(\phi(\hat{\boldsymbol{z}}_{d+1}) - \boldsymbol{x}_{d+1})^\top \frac{\partial \phi(\hat{\boldsymbol{z}}_{d+1})}{\partial w_{j,d+1}} = \frac{\sigma_j(\boldsymbol{w}_{d+1})}{n} \langle \phi(\hat{\boldsymbol{z}}_{d+1}) - \boldsymbol{x}_{d+1}, \phi'(\hat{\boldsymbol{z}}_{d+1}), \boldsymbol{x}_j - \hat{\boldsymbol{z}}_{d+1} \rangle \tag{13}$$

$$= -\frac{1}{nd} \langle \boldsymbol{x}_{d+1}, -2c\hat{\boldsymbol{z}}_{d+1}, \boldsymbol{x}_j - \hat{\boldsymbol{z}}_{d+1} \rangle \tag{14}$$

$$+ \frac{1}{nd} \langle -\mathbf{1}_n + c\hat{\boldsymbol{z}}_{d+1}^2, 2c\hat{\boldsymbol{z}}_{d+1}, \boldsymbol{x}_j - \hat{\boldsymbol{z}}_{d+1} \rangle \tag{15}$$

$$+ \frac{1}{nd} \langle O(|\hat{\boldsymbol{z}}_{d+1}|^4), -2c\hat{\boldsymbol{z}}_{d+1}, \boldsymbol{x}_j - \hat{\boldsymbol{z}}_{d+1} \rangle \tag{16}$$

$$+ \frac{1}{nd} \langle \phi(\hat{\boldsymbol{z}}_{d+1}) - \boldsymbol{x}_{d+1}, O(|\hat{\boldsymbol{z}}_{d+1}|^3), \boldsymbol{x}_j - \hat{\boldsymbol{z}}_{d+1} \rangle. \tag{17}$$

Like the original proof, we will show that the first term is the leading term and the other three vanish eventually. Recall that under the CoT setting, thanks to the binary tree structure, the multi-linear product among a parent and two children is always one.

In our case, however, this no longer holds: Suppose $d = 16$, $k = 4$, $x_{17}$ is the root, and $x_2, x_3, x_6, x_7$ are relevant bits. Observe that the product $\langle x_{17}, x_2, x_7 \rangle = x_2 x_3 x_6 x_7 x_2 x_7 = x_3 x_6$ is a random variable instead of a fixed value.

Nevertheless, we will see the random variables are largely homogeneous. We first express the leading term in a more readable way by substituting $\hat{\boldsymbol{z}}_{d+1} = \frac{1}{d}\sum_\alpha \boldsymbol{x}_\alpha$. Observe that,

$$\frac{1}{n} \langle \boldsymbol{x}_{d+1}, \hat{\boldsymbol{z}}_{d+1}, \boldsymbol{x}_j - \hat{\boldsymbol{z}}_{d+1} \rangle = \frac{1}{nd} \sum_\alpha \langle \boldsymbol{x}_{d+1}, \boldsymbol{x}_\alpha, \boldsymbol{x}_j \rangle - \frac{1}{nd^2} \sum_{\alpha,\beta} \langle \boldsymbol{x}_{d+1}, \boldsymbol{x}_\alpha, \boldsymbol{x}_\beta \rangle, \tag{18}$$

where the dummy indices $\alpha$ and $\beta$ are taken to run over $[d]$. Before continuing, we prove a lemma that provides a concentration bound for the interactions between the bits.

**Lemma B.1.** *Using the distribution $\mathcal{D}$, and let $r \in [4]$, then for each of the following two cases:*

    *1. $r$ is odd*

    *2. $r$ is even but at least one of $j_1, \ldots, j_r$ is an irrelevant bit.*

*Then for any $p > 0$, it holds with probability at least $1 - p$ that*

$$\max_{r \in [4], \{j_1, \ldots, j_r \not\subseteq P\}} \frac{|\langle \boldsymbol{x}_{j_1}, \ldots, \boldsymbol{x}_{j_r} \rangle|}{n} \leq \kappa := \sqrt{\frac{2}{n} \log \frac{4d^4}{p}}. \tag{19}$$

*Otherwise, if the indices of $r$ bits are not any one of the two case above, then similarly, for any $p > 0$, it holds with probability as least $1 - p$ that*

$$\max_{r \in [4], \{j_1, \ldots, j_r \not\subseteq P\}} \frac{|\langle \boldsymbol{x}_{j_1}, \ldots, \boldsymbol{x}_{j_r} \rangle - (1 - \rho)|}{n} \leq \kappa := \sqrt{\frac{2}{n} \log \frac{4d^4}{p}}. \tag{20}$$

*Proof.* We prove the cases when $r$ is odd or $r$ is even but at least one of $j_1, \ldots, j_r$ is an irrelevant bit. The proof for the remaining scenario with non-zero mean is the same. Observe that, for each sample, the multi-linear product $\langle x_{j_1}, \ldots, x_{j_r} \rangle = x_{j_1} \cdots x_{j_r}$ is a random variable. Suppose $r$ is odd, i.e. $r = 1$ or $3$ and the bits are all relevant. Let $j \in P$, then $\mathbb{P}_{\mathcal{D}}(x_j = 1) = \rho \times 0.5 + (1 - \rho) \times 0.5 = 0.5 = \mathbb{P}_{\mathcal{D}}(x_j = -1)$, so $\mathbb{E}_{\mathcal{D}}[x_j] = 0$. Let $j_1, j_2, j_3 \in P$, we have

$$\mathbb{P}_{\mathcal{D}}(x_{j_1} x_{j_2} x_{j_3} = 1) = \rho \times \frac{\binom{3}{1} + \binom{3}{3}}{2^3} + (1 - \rho) \times \frac{1}{2} = 0.5 = \mathbb{P}_{\mathcal{D}}(x_{j_1} x_{j_2} x_{j_3} = -1). \tag{21}$$

Therefore, $\mathbb{E}_{\mathcal{D}}[x_{j_1} x_{j_2} x_{j_3}] = 0$.

Next, suppose $j_1 \notin P$, then clearly $\mathbb{P}_{\mathcal{D}}[x_{j_1} = 1] = \mathbb{P}_{\mathcal{D}}[x_{j_1} = -1] = \rho \times 0.5 + (1 - \rho) \times 0.5 = 0.5$, so $\mathbb{E}_{\mathcal{D}}[x_{j_1}] = \mathbb{E}_{\text{Uniform}}[x_{j_1}] = 0$. Non-relevant bits are independent with respect to every other bit, so $\mathbb{E}_{\mathcal{D}}[\langle x_{j_1}, \ldots, x_{j_r} \rangle] = \mathbb{E}_{\text{Uniform}}[\langle x_{j_1}, \ldots, x_{j_r} \rangle] = 0$.

Since all the variables discussed above have zero mean, by Hoeffding's inequality, we have

$$\mathbb{P}_{\mathcal{D}}\left(|\langle x_{j_1}, \ldots, x_{j_r} \rangle| \geq \lambda\right) \leq 2e^{-\lambda^2/2n}. \tag{22}$$

If $r = 1$, there are exactly $d - k$ such variables; if $r = 2$, there are $d(k-1)$; if $r = 3$, there are $d(k-1)(k-2)$; if $r = 4$, there are $d(k-1)(k-2)(k-3)$. Their sum is below $2d^4$. So, by union bounding, we have

$$\mathbb{P}\left(\max_{r \in [4], \{j_1, \ldots, j_r \not\subseteq P\}} |\langle \boldsymbol{x}_{j_1}, \ldots, \boldsymbol{x}_{j_r} \rangle| \geq \lambda\right) \leq 4d^4 e^{-\lambda^2/2n}. \tag{23}$$

The lemma statement follows by substituting $\lambda = \sqrt{\frac{2}{n} \log \frac{4d^4}{p}}$. $\qquad\square$

If we take $n = \Omega(d^{2+\epsilon})$ and $p = \exp\left(-d^{\epsilon/2}\right)$, so $\kappa = O\left(d^{-1-\epsilon/4}\right)$.

### B.1 Term (15)

We now proceed to analyze the quantity of the leading term when $j \in P$, i.e. $j$ is a relevant bit. Thus, we can decompose the two terms in Equation (18) as:

$$\frac{1}{nd} \sum_{\alpha} \langle \boldsymbol{x}_{d+1}, \boldsymbol{x}_{\alpha}, \boldsymbol{x}_j \rangle = \frac{1}{nd} \sum_{\alpha \in P} \langle \boldsymbol{x}_{d+1}, \boldsymbol{x}_{\alpha}, \boldsymbol{x}_j \rangle + \frac{1}{nd} \sum_{\alpha \notin P} \langle \boldsymbol{x}_{d+1}, \boldsymbol{x}_{\alpha}, \boldsymbol{x}_j \rangle \tag{24}$$

$$= \frac{1}{nd} \sum_{\alpha \in P} \langle \boldsymbol{x}_{d+1}, \boldsymbol{x}_{\alpha}, \boldsymbol{x}_j \rangle + \frac{1}{d} \cdot O((d-k)\kappa) \tag{25}$$

$$= \frac{1}{d} \cdot X_k + \frac{k-1}{d} \cdot X_{k-2} + \frac{d-k}{d} O(\kappa). \tag{26}$$

Similarly,

$$\frac{1}{nd^2} \sum_{\alpha} \langle \boldsymbol{x}_{d+1}, \boldsymbol{x}_{\alpha}, \boldsymbol{x}_{\beta} \rangle = \frac{1}{nd^2} \sum_{\alpha,\beta \in P} \langle \boldsymbol{x}_{d+1}, \boldsymbol{x}_{\alpha}, \boldsymbol{x}_{\beta} \rangle + \frac{1}{nd^2} \sum_{\text{rest}} \langle \boldsymbol{x}_{d+1}, \boldsymbol{x}_{\alpha'}, \boldsymbol{x}_{\beta'} \rangle \tag{27}$$

$$= \frac{1}{nd^2} \sum_{\alpha,\beta \in P} \langle \boldsymbol{x}_{d+1}, \boldsymbol{x}_{\alpha}, \boldsymbol{x}_{\beta} \rangle + O\left( \frac{d^2 - k^2}{d^2} \cdot \kappa \right) \tag{28}$$

$$= \frac{k}{d^2} \cdot X_k + \frac{k(k-1)}{d^2} \cdot X_{k-2} + \frac{d^2 - k^2}{d^2} O(\kappa). \tag{29}$$

Adding them up, we have that, if $j \in P$, then

$$\frac{1}{n} \langle \boldsymbol{x}_{d+1}, \hat{\boldsymbol{z}}_{d+1}, \boldsymbol{x}_j - \hat{\boldsymbol{z}}_{d+1} \rangle = \frac{d-k}{d^2} X_k + \frac{(d-k)(k-1)}{d^2} X_{k-2} - \frac{k(d-k)}{d^2} O(\kappa). \tag{30}$$

Multiplying $\frac{2c}{d} = \frac{2ak^2}{d}$, we finally have

$$\frac{2c}{nd} \langle \boldsymbol{x}_{d+1}, \hat{\boldsymbol{z}}_{d+1}, \boldsymbol{x}_j - \hat{\boldsymbol{z}}_{d+1} \rangle = \frac{2ak^2(d-k)}{d^3} X_k + \frac{2ak^2(d-k)(k-1)}{d^3} X_{k-2} - \frac{2ak^3(d-k)}{d^3} O(\kappa). \tag{31}$$

On the other hand, if $j \notin P$, then we have

$$\frac{1}{n} \langle \boldsymbol{x}_{d+1}, \hat{\boldsymbol{z}}_m, \boldsymbol{x}_j \rangle = \frac{1}{nd} \sum_{\alpha} \langle \boldsymbol{x}_{d+1}, \boldsymbol{x}_{\alpha}, \boldsymbol{x}_j \rangle - \frac{1}{nd^2} \sum_{\alpha,\beta} \langle \boldsymbol{x}_{d+1}, \boldsymbol{x}_{\alpha}, \boldsymbol{x}_{\beta} \rangle \tag{32}$$

$$= \frac{1}{d} X_k + \frac{d-1}{d} O(\kappa) - \left( \frac{k}{d^2} X_k + \frac{k(k-1)}{d^2} X_{k-2} + \frac{d^2 - k^2}{d^2} O(\kappa) \right) \tag{33}$$

$$= \frac{d-k}{d^2} X_k - \frac{k(k-1)}{d^2} X_{k-2} + \frac{k^2 - d}{d^2} O(\kappa). \tag{34}$$

Again, multiplying $\frac{2c}{d} = \frac{2ak^2}{d}$, we have

$$\frac{2c}{nd} \langle \boldsymbol{x}_{d+1}, \hat{\boldsymbol{z}}_m, \boldsymbol{x}_j \rangle = \frac{2ak^2(d-k)}{d^3} X_k - \frac{2ak^3(k-1)}{d^3} X_{k-2} + \frac{2ak^2(k^2 - d)}{d^3} O(\kappa). \tag{35}$$

Therefore, we have

$$\text{Term (15)} = \begin{cases} \frac{2ak^2(d-k)}{d^3} X_k + \frac{2ak^2(d-k)(k-1)}{d^3} X_{k-2} - \frac{2ak^3(d-k)}{d^3} O(\kappa), & j \in P \\ \frac{2ak^2(d-k)}{d^3} X_k - \frac{2ak^3(k-1)}{d^3} X_{k-2} + \frac{2ak^2(k^2-d)}{d^3} O(\kappa), & j \notin P \end{cases}. \tag{36}$$

### B.2 Term (16)

We expand term (16) as the following:

$$\frac{1}{nd} \langle -\boldsymbol{1}_n + c\hat{\boldsymbol{z}}_{d+1}^2, 2c\hat{\boldsymbol{z}}_{d+1}, \boldsymbol{x}_j - \hat{\boldsymbol{z}}_{d+1} \rangle = -\frac{2c}{nd} \langle \hat{\boldsymbol{z}}_{d+1}, \boldsymbol{x}_j \rangle + \frac{2c}{nd} \langle \hat{\boldsymbol{z}}_{d+1}^2 \rangle + \frac{2c^2}{nd} \langle \hat{\boldsymbol{z}}_{d+1}^3, \boldsymbol{x}_j \rangle - \frac{2c^2}{nd} \langle \hat{\boldsymbol{z}}_{d+1}^4 \rangle. \tag{37}$$

### B.2.1 First term

$$\frac{1}{n}\langle \hat{\boldsymbol{z}}_{d+1}, \boldsymbol{x}_j \rangle = \frac{1}{nd}\left(\langle \boldsymbol{x}_j, \boldsymbol{x}_j \rangle + \sum_{\alpha \neq j}\langle \boldsymbol{x}_\alpha, \boldsymbol{x}_j \rangle\right) = \frac{1}{d} + \frac{1}{nd}\sum_{\alpha \neq j}\langle \boldsymbol{x}_\alpha, \boldsymbol{x}_j \rangle \tag{38}$$

$$= \begin{cases} 1/d + \frac{1}{nd}\sum_{\alpha \in P \setminus \{j\}}\langle \boldsymbol{x}_\alpha, \boldsymbol{x}_j \rangle, & j \in P \\ 1/d + \frac{d-1}{d}O(\kappa), & j \notin P \end{cases} \tag{39}$$

$$= \begin{cases} 1/d + \frac{k-1}{d}X_2 + \frac{d-k}{d}O(\kappa), & j \in P \\ 1/d + \frac{d-1}{d}O(\kappa), & j \notin P \end{cases} \tag{40}$$

Multiplying the constant $-2c$, we have

$$-\frac{2c}{nd}\langle \hat{\boldsymbol{z}}_{d+1}, \boldsymbol{x}_j \rangle = \begin{cases} -\frac{2c}{d^2} - \frac{2c(k-1)}{d^2}X_2 - \frac{2c(d-k)}{d^2}O(\kappa), & j \in P \\ -\frac{2c}{d^2} - \frac{2c(d-1)}{d^2}O(\kappa), & j \notin P \end{cases} \tag{41}$$

$$= \begin{cases} -\frac{2ak^2}{d^2} - \frac{2ak^2(k-1)}{d^2}X_2 - \frac{2ak^2(d-k)}{d^2}O(\kappa), & j \in P \\ -\frac{2ak^2}{d^2} - \frac{2ak^2(d-1)}{d^2}O(\kappa), & j \notin P \end{cases} \tag{42}$$

### B.2.2 Second term

Next,

$$\frac{1}{n}\langle \hat{\boldsymbol{z}}_{d+1}^2 \rangle = \frac{1}{nd^2}\left(\sum_\alpha \langle \boldsymbol{x}_\alpha, \boldsymbol{x}_\alpha \rangle + \sum_{\alpha \neq \beta}\langle \boldsymbol{x}_\alpha, \boldsymbol{x}_\beta \rangle\right) \tag{43}$$

$$= \frac{1}{d} + \frac{1}{nd^2}\sum_{\alpha \neq \beta; \alpha, \beta \in P}\langle \boldsymbol{x}_\alpha, \boldsymbol{x}_\beta \rangle + \frac{1}{nd^2}\sum_{\text{rest}}\langle \boldsymbol{x}_\alpha, \boldsymbol{x}_\beta \rangle \tag{44}$$

$$= \frac{1}{d} + \frac{k(k-1)}{d^2}X_2 + \frac{(d+k-1)(d-k)}{d^2}O(\kappa) \tag{45}$$

Therefore,

$$\frac{2c}{nd}\langle \hat{\boldsymbol{z}}_{d+1}^2 \rangle = \frac{2c}{d^2} + \frac{2ck(k-1)}{d^3}X_2 + \frac{2c(d+k-1)(d-k)}{d^3}O(\kappa) \tag{46}$$

$$= \frac{2ak^2}{d^2} + \frac{2ak^3(k-1)}{d^3}X_2 + \frac{2ak^2(d+k-1)(d-k)}{d^3}O(\kappa). \tag{47}$$

### B.2.3 Fourth term

For the fourth order term, we have

$$\frac{1}{n}\langle \hat{\boldsymbol{z}}_{d+1}^4 \rangle = \frac{1}{nd^4}\sum_{\alpha, \beta, \gamma, \delta}\langle \boldsymbol{x}_\alpha, \boldsymbol{x}_\beta, \boldsymbol{x}_\gamma, \boldsymbol{x}_\delta \rangle \tag{48}$$

We analyze all possible combinations of the indices by enumerating the size of the set $\{\alpha, \beta, \gamma, \delta\}$.

1. $|\{\alpha, \beta, \gamma, \delta\}| = 1$: There are $d$ instances of $\langle \boldsymbol{x}_\alpha, \boldsymbol{x}_\alpha, \boldsymbol{x}_\alpha, \boldsymbol{x}_\alpha \rangle$.
2. $|\{\alpha, \beta, \gamma, \delta\}| = 2$: There are $\binom{d}{2}$ pairs of distinct indices. For each pair $(\alpha, \beta)$, either:
   (a) One appears three times. There are eight possibilities: We first choose the position of the unique index (four), and both can be that unique index (multiplying two).
   (b) Both appear twice. Clearly, there are six possibilities.
   Therefore, there are totally $14\binom{d}{2} = 7d(d-1)$ possible combinations in this case.

3. $|\{\alpha, \beta, \gamma, \delta\}| = 3$: Given a triple $(\alpha, \beta, \gamma)$, exactly one index must appear twice, so there are three possibilities for this criteria. Once this choice is made, the identical indices may choose one of the six pairs among the four positions. Once two spots are occupied, the remaining two indices may fill in either order. So, there are totally $36\binom{d}{3} = 6d(d-1)(d-2)$ possible combinations in this case.

4. $|\{\alpha, \beta, \gamma, \delta\}| = 4$: Clearly, there are $\binom{d}{4}$ groups of all distinct indices. For each group, they may be placed in $4! = 24$ possible orders. So there are totally $24\binom{d}{4} = d(d-1)(d-2)(d-3)$ possible combinations in this case.

Every instance in Cases 1 and 2.b has value exactly $n$, and there are $d + 6\binom{d}{2} = 3d^2 - 2d$ such instances. Every instances in Cases 2.a and 3 is a sum of $n$ independent (data samples) random variables, and each of them is in the form $\langle x_\alpha, x_\beta \rangle$ such that $\alpha \neq \beta$. Every instance in Case 4 is a sum of $n$ independent random variables in the form $\langle x_\alpha, x_\beta, x_\gamma, x_\delta \rangle$ such that $|\{\alpha, \beta, \gamma, \delta\}| = 4$. So we further divide those $d^4$ instance into three groups:

- Cases 1 and 2.b.
- Cases 2.a and 3. Among $6d^3 - 14d^2 + 8d$ such instances, exactly $8\binom{k}{2} + 36\binom{k}{3} = 6k^3 - 14k^2 + 8k$ correspond to random variables as multiplication of two bits in $P$. The rest of them have value $O(\kappa)$.
- Case 4. Clearly $24\binom{k}{4} = k(k-1)(k-2)(k-3)$ instances correspond to random variables as multiplication of four bits in $P$. The rest of them have value $O(\kappa)$.

The first group with $3d^2 - 2d$ instances accumulates to

$$\frac{1}{nd^4} \times (\text{Cases 1 and 2.b}) = \frac{1}{nd^4} \times n(3d^2 - 2d) = \frac{3d - 2}{d^3}. \tag{49}$$

The second group leads to

$$\frac{1}{nd^4} \times (\text{Cases 2.a and 3}) = \frac{6k^3 - 14k^2 + 8k}{d^4} \cdot X_2 + \frac{6(d^3 - k^3) - 14(d^2 - k^2) + 8(d - k)}{d^4} O(\kappa). \tag{50}$$

The third group leads to

$$\frac{1}{nd^4} \times (\text{Case 4}) = \frac{k(k-1)(k-2)(k-3)}{d^4} \cdot X_4 + \frac{d(d-1)(d-2)(d-3) - k(k-1)(k-2)(k-3)}{d^4} O(\kappa). \tag{51}$$

Adding everything up, we have

$$\frac{1}{n} \langle \hat{z}_{d+1}^4 \rangle = \frac{3d - 2}{d^3} + \frac{6k^3 - 14k^2 + 8k}{d^4} \cdot X_2 + \frac{k(k-1)(k-2)(k-3)}{d^4} \cdot X_4 \tag{52}$$

$$+ \frac{(d^4 - k^4) - 3(d^2 - k^2) + 2(d - k)}{d^4} O(\kappa). \tag{53}$$

Multiplying $-2c^2 = -2a^2k^4$, we have

$$-\frac{2c^2}{nd} \langle \hat{z}_{d+1}^4 \rangle = -\frac{2a^2k^4(3d^2 - 2d)}{d^5} - \frac{2a^2k^4(6k^3 - 14k^2 + 8k)}{d^5} \cdot X_2 - \frac{2a^2k^5(k-1)(k-2)(k-3)}{d^5} \cdot X_4 \tag{54}$$

$$- \frac{2a^2k^4\left[(d^4 - k^4) - 3(d^2 - k^2) + 2(d - k)\right]}{d^5} \cdot O(\kappa). \tag{55}$$

### B.2.4 Third term

Finally, for the three order term, where the group $\{\alpha, \beta, \gamma, \delta\}$ must contain a fixed $j \in [d]$. We have,

$$\frac{1}{n} \langle \hat{z}_{d+1}^3, x_j \rangle = \frac{1}{nd^3} \sum_{\alpha, \beta, \gamma} \langle x_\alpha, x_\beta, x_\gamma, x_j \rangle \tag{56}$$

Like the previous four order term, we need to discuss multiple cases.

1. $\alpha = \beta = \gamma = j$. We easily obtain $\frac{1}{nd^3}\langle x_j, x_j, x_j, x_j \rangle = 1/d^3$.

2. $\alpha = \beta = \gamma \neq j$. There are $d - 1$ possible combinations.

3. $|\{\alpha, \beta, \gamma\}| = |\{\alpha, \beta, \gamma, j\}| = 2$. This further divides into two sub-cases.

    (a) Only one of them is $j$. We have $d - 1$ choices for $\alpha \neq j$, and three choices of spot for that unique $j$. Totally, there are $3(d - 1)$ combinations.

    (b) Two of them are $j$. Like the sub-case above, there are $3(d - 1)$ possible combinations.

4. $|\{\alpha, \beta, \gamma\}| = 2$, but $|\{\alpha, \beta, \gamma, j\}| = 3$. None of the three indices can be $j$, so there are $\binom{d-1}{2}$ pairs of $(\alpha, \beta)$. For each pair, we need to choose which index appears once between two elements. In total, there are $6\binom{d-1}{2} = 3(d - 1)(d - 2)$ combinations.

5. $|\{\alpha, \beta, \gamma\}| = |\{\alpha, \beta, \gamma, j\}| = 3$. Exactly one of the three indices must be $j$, so there are $\binom{d-1}{2}$ pairs of unequal indices. There are three choices for $j$'s position, and every time we can flip the pair. Therefore, there are $6\binom{d-1}{2} = 3(d - 1)(d - 2)$ combinations in this case.

6. $|\{\alpha, \beta, \gamma\}| = 3$, and $|\{\alpha, \beta, \gamma, j\}| = 4$. There are $\binom{d-1}{3}$ triples $(\alpha, \beta, \gamma)$ and six combinations for each choice, so there are $6\binom{d-1}{3} = (d - 1)(d - 2)(d - 3)$ combinations in total.

The unique case in Case 1 is trivial.

We first assume $j \notin P$. Then Lemma B.1 directly applies to all summands and therefore the term sums to the following with high probability:

$$\frac{2c^2}{nd}\langle \hat{z}_{d+1}^3, x_j \rangle = \frac{2c^2}{nd^4}\sum_{\alpha,\beta,\gamma}\langle x_\alpha, x_\beta, x_\gamma, x_j \rangle = \frac{2c^2 \cdot \langle x_\alpha, x_\beta, x_\gamma, x_j \rangle}{nd^4} \times d^3 = \frac{2c^2}{d} \cdot O(\kappa) = \frac{2a^2 k^4}{d}O(\kappa). \quad (57)$$

Now suppose $j \in P$. Exactly $k - 1$ combinations in Case 2 correspond to a sum of $n$ independent $X_2$, and the remaining $d - k$ combinations are $O(\kappa)$. All instances in Case 3.a have product value $n$, and all instances in Case 3.b correspond to $n$ copies of $X_2$. For Case 4, only $j$ and the unique index matter. Exactly $2(k-1)(d-2)$ combinations correspond to $n$ copies of $X_2$, and the rest are $O(\kappa)$. For Case 5, the $x_j$ cancels out, so the sum for Case 5 can be expressed as

$$\sum_{\alpha \neq \beta; \alpha, \beta \in [d]\backslash\{j\}} \langle x_\alpha, x_\beta \rangle. \quad (58)$$

Clearly, only $3(k - 1)(k - 2)$ summands correspond to $n$ copies of $X_2$, and others are $O(\kappa)$. Finally, for Case 6, there are $6\binom{k-1}{3}$ combinations that correspond to $n$ copies of $X_4$, and all others are $O(\kappa)$. We can then express the three order term as the following.

$$\frac{1}{n}\langle \hat{z}_{d+1}^3, x_j \rangle = \frac{1}{nd^3}\sum_{\alpha,\beta,\gamma}\langle x_\alpha, x_\beta, x_\gamma, x_j \rangle \quad (59)$$

$$= \frac{3d - 2}{d^3} + \frac{(k - 1) + 3(d - 1) + 2(k - 1)(d - 2) + 3(k - 1)(k - 2)}{d^3} \cdot X_2 \quad (60)$$

$$+ \frac{(k - 1)(k - 2)(k - 3)}{d^3} \cdot X_4 \quad (61)$$

$$+ \frac{(d^3 - k^3) - 2dk + 3k^2 - 4d + k + 2}{d^3} \cdot O(\kappa). \quad (62)$$

Multiplying $2c^2 = 2a^2 k^4$, we have

$$\frac{2c^2}{nd}\langle \hat{z}_{d+1}^3, x_j \rangle = \frac{1}{nd^4}\sum_{\alpha,\beta,\gamma}\langle x_\alpha, x_\beta, x_\gamma, x_j \rangle \quad (63)$$

$$= \frac{2a^2 k^4(3d - 2)}{d^4} + \frac{2a^2 k^4\left[(k - 1) + 3(d - 1) + 2(k - 1)(d - 2) + 3(k - 1)(k - 2)\right]}{d^4} \cdot X_2 \quad (64)$$

$$+ \frac{2a^2 k^4(k - 1)(k - 2)(k - 3)}{d^4} \cdot X_4 \quad (65)$$

$$+ \frac{2a^2 k^4\left[(d^3 - k^3) - 2dk + 3k^2 - 4d + k + 2\right]}{d^4} \cdot O(\kappa). \quad (66)$$

### B.2.5 Adding everything up

Adding everything up, we have

$$\text{Term (16)} = -\frac{2c}{nd} \langle \hat{\boldsymbol{z}}_{d+1}, \boldsymbol{x}_j \rangle + \frac{2c}{nd} \langle \hat{\boldsymbol{z}}_{d+1}^2 \rangle + \frac{2c^2}{nd} \langle \hat{\boldsymbol{z}}_{d+1}^3, \boldsymbol{x}_j \rangle - \frac{2c^2}{nd} \langle \hat{\boldsymbol{z}}_{d+1}^4 \rangle. \tag{67}$$

Specifically,

- If $j \in P$, then
  - The coefficient for constant is 0.
  - The coefficient for $X_2$ is

$$\frac{(4a^2 \cdot d^2 k^5 + 6a^2 \cdot dk^6 - 12a^2 \cdot k^7) + (-2a \cdot d^3 k^3 + 2a^2 \cdot d^2 k^4 - 24a^2 \cdot dk^5 + 28a^2 \cdot k^6)}{d^5} \tag{68}$$

$$+ \frac{(2a \cdot d^3 k^2 - 2a \cdot d^2 k^3 + 12a^2 \cdot dk^4 - 16a^2 k^5)}{d^5} = O(d^2). \tag{69}$$

  - The coefficient for $X_4$ is

$$\frac{2ak^4(d-k)(k-1)(k-2)(k-3)}{d^5} \geq 0; \quad \Rightarrow O(d^3). \tag{70}$$

  - The coefficient for $O(\kappa)$ is

$$\frac{-4a^2 d^2 k^5 - 2a^2 d^2 k^4 - 2a^2 dk^7 + 6a^2 dk^6 + 2a^2 dk^5 + 2a^2 k^8 - 6a^2 k^6 + 4a^2 k^5}{d^5} \tag{71}$$

$$+ \frac{2ad^3 k^3 - 2ad^3 k^2 - 2ad^2 k^3 + 2ad^2 k^2}{d^5} = -O(d^2). \tag{72}$$

  Hence, this term becomes $-O(d^{1-\epsilon/4})$.
- If $j \notin P$, then
  - The coefficient for constant is

$$-\frac{2a^2 k^4(3d-2)}{d^4} = -O(d). \tag{73}$$

  - The coefficient for $X_2$ is

$$\frac{-12a^2 \cdot k^7 + 2a \cdot d^2 k^4 + 28a^2 \cdot k^6 - 2a \cdot d^2 k^3 - 16a^2 \cdot k^5}{d^5} = -O(d^2). \tag{74}$$

  - The coefficient for $X_4$ is

$$-\frac{2a^2 k^5(k-1)(k-2)(k-3)}{d^5} = -O(d^3). \tag{75}$$

  - The coefficient for $O(\kappa)$ is

$$\frac{6a^2 d^2 k^4 - 4a^2 dk^4 + 2a^2 k^7 - 6a^2 k^6 + 4a^2 k^5 - 2ad^2 k^4 + 2ad^2 k^3}{d^5} = O(d^2). \tag{76}$$

  Hence, this term becomes $O(d^{1-\epsilon/4})$.

### B.3 Terms (17) and (18)

Now we analyze term (17). Recall that $\left\langle |\hat{\boldsymbol{z}}_{d+1}|^4 \right\rangle = \left\langle \hat{\boldsymbol{z}}_{d+1}^4 \right\rangle = O(n/d^2) + O(n/d) \cdot X_2 + O(n) \cdot X_4 + O(n\kappa)$. Observe that each component of $\hat{\boldsymbol{z}}_{d+1}$ and $\boldsymbol{x}_j - \hat{\boldsymbol{z}}_{d+1}$ are contained in $[-1, 1]$ and $[-2, 2]$ respectively, we have

$$\frac{1}{nd} \left\langle O(|\hat{\boldsymbol{z}}_{d+1}|^4), -2c\hat{\boldsymbol{z}}_{d+1}, \boldsymbol{x}_j - \hat{\boldsymbol{z}}_{d+1} \right\rangle = -\frac{4c}{nd} \cdot O\left( \left\langle |\hat{\boldsymbol{z}}_{d+1}|^4 \right\rangle \right) \tag{77}$$

$$= -\frac{4ak^2}{d} \left( \frac{3d-2}{d^3} + \frac{6k^3 - 14k^2 + 8k}{d^4} \cdot X_2 + \frac{k(k-1)(k-2)(k-3)}{d^4} \cdot X_4 + O(d^{-2-\epsilon/4}) \right) \tag{78}$$

$$= -O(d^{-1}) - O(1) \cdot X_2 - O(d) \cdot X_4 - O(\kappa). \tag{79}$$

Similarly, using the Cauchy-Schwarz inequality, we may bound the final term (18).

$$\frac{1}{nd}\left\langle \phi(\hat{\boldsymbol{z}}_{d+1}) - \boldsymbol{x}_{d+1}, O(|\hat{\boldsymbol{z}}_{d+1}|^3), \boldsymbol{x}_j - \hat{\boldsymbol{z}}_{d+1} \right\rangle = \frac{4}{nd} O\left(\langle|\hat{\boldsymbol{z}}_{d+1}|\rangle^3\right) \tag{80}$$

$$\leq \frac{4}{nd}\left\langle \hat{\boldsymbol{z}}_{d+1}^2 \right\rangle^{1/2}\left\langle \hat{\boldsymbol{z}}_{d+1}^4 \right\rangle^{1/2} \tag{81}$$

$$\leq O\left(d^{1+0.5\epsilon}\right) \tag{82}$$

Now, summing up the terms, we conclude that for relevant bits $j \in P$, its gradient $\partial L/\partial w_{d+1,j}$ has the dominating term $O(d^3) \cdot X_4 = O(d^3)$; for non-relevant bits $j' \notin P$, its gradient $\partial L/\partial w_{d+1,j'}$ has the dominating term $O(d^{1-\epsilon/4})$. Fix a learning rate $\eta = \Theta(d^{-3+\epsilon/8})$, then we obtain the following comparisons of the weights $\mathbf{W}^{(1)}$ after one gradient update. For $j \in P$ and $j' \notin P$, we have

$$\frac{\sigma_{j'}(w_{d+1}^{(1)})}{\sigma_j(w_{d+1}^{(1)})} = e^{w_{d+1,j'}^{(1)} - w_{d+1,j}^{(1)}} \leq \exp\left(-\Omega(d^{\epsilon/8})\right). \tag{83}$$

Since attention scores sum up to one, we have $\sum_{j \in P} \sigma_j(\boldsymbol{w}^{(1)})$ If both $j, k \in P$, then the higher order terms cancel out and the perturbation terms for correct gradient updates become $O(d^{-\epsilon/4})$. Therefore, we have

$$\frac{\sigma_j(w_{d+1}^{(1)})}{\sigma_k(w_{d+1}^{(1)})} = \frac{\sigma_k(w_{d+1}^{(1)})}{\sigma_j(w_{d+1}^{(1)})} \leq \exp(O(d^{-2-\epsilon/8})) \leq 1 + O(d^{-2-\epsilon/8}), \tag{84}$$

where the last inequality holds because $e^t \leq 1 + O(t)$ for small $t > 0$. The ratio holds for all $k$ elements in $P$, so for any $j \in P$, we have

$$\frac{1}{k} - O(d^{-2-\epsilon/8}) \leq \sigma_j(\boldsymbol{w}^{(1)}) \leq \frac{1}{k} + O(d^{-2-\epsilon/8}). \tag{85}$$

Therefore, for any $d$-dimensional input $\boldsymbol{x}$, the prediction $\hat{y} = \hat{x}_{d+1}$ satisfies the following inequality:

$$|y - \hat{y}| = |\phi(\hat{z}_{d+1}) - \phi(z_{d+1})| \tag{86}$$

$$\leq k \times |\hat{z}_{d+1} - z_{d+1}| \tag{87}$$

$$= k \times \left|O(d^{-2-\epsilon/8}) + (d - k) \times \exp\left(-\Omega(d^{\epsilon/8})\right)\right| \tag{88}$$

$$= k \cdot O(d^{-2-\epsilon/8}) = O(d^{-1-\epsilon/8}). \tag{89}$$

## C   Proof of Theorem 4.2

The high-level ideas are identical with the previous proof. We still apply Lemma B.1 for this proof to bound the perturbation terms. To reach the conclusion, we must compute the derivative of loss with respect to weights:

$$\frac{\partial L}{\partial w_{j,m}}(\mathbf{W}) = \frac{1}{n(m-1)}\left\langle \phi(\hat{\boldsymbol{z}}_m - \boldsymbol{x}_m), \phi'(\hat{\boldsymbol{z}}_m), \boldsymbol{x}_j - \hat{\boldsymbol{z}}_m \right\rangle \tag{90}$$

$$= \frac{1}{n(m-1)}\left\langle 2\hat{\boldsymbol{z}}_m - 1 - \boldsymbol{x}_m, \phi'(\hat{\boldsymbol{z}}_m), \boldsymbol{x}_j - \hat{\boldsymbol{z}}_m \right\rangle. \tag{91}$$

We can write $\phi'(\hat{z}_m) = 2d^3 \hat{z}_m$, so the derivative becomes

$$\frac{\partial L}{\partial w_{j,m}}(\mathbf{W}) = \frac{1}{n(m-1)}\left\langle 2\hat{\boldsymbol{z}}_m - 1 - \boldsymbol{x}_m, 2d^3 \hat{\boldsymbol{z}}_m, \boldsymbol{x}_j - \hat{\boldsymbol{z}}_m \right\rangle \tag{92}$$

$$= -\frac{1}{n(m-1)}\left\langle \boldsymbol{x}_m, 2d^3 \hat{\boldsymbol{z}}_m, \boldsymbol{x}_j - \hat{\boldsymbol{z}}_m \right\rangle \tag{93}$$

$$+ \frac{1}{n(m-1)}\left\langle -\mathbf{1}_n + 2\hat{\boldsymbol{z}}_m, 2d^3 \hat{\boldsymbol{z}}_m, \boldsymbol{x}_j - \hat{\boldsymbol{z}}_m \right\rangle \tag{94}$$

The quadratic derivative only holds in $[-d^{-3}, d^{-3}]$, and we will bound it asymptotically, we may replace it with 2 here.

The structure of the proof is then divided into three parts.

- Sections B.1-B.3 are computations of the gradients and gradient differences that lead to conditions in Equation (7) and (8).
- Section B.4 is the *if* direction of Theorem 4.2.
- Section B.5 is the *only if* direction of the theorem.

### C.1  The first term

We first rewrite the first term as

$$-\frac{2}{n(m-1)}\langle \boldsymbol{x}_m, \hat{\boldsymbol{z}}_m, \boldsymbol{x}_j - \hat{\boldsymbol{z}}_m\rangle = -\frac{2}{n(m-1)^2}\sum_{\alpha}\langle \boldsymbol{x}_m, \boldsymbol{x}_\alpha, \boldsymbol{x}_j\rangle + \frac{2}{n(m-1)^2}\sum_{\alpha,\beta}\langle \boldsymbol{x}_m, \boldsymbol{x}_\alpha, \boldsymbol{x}_\beta\rangle. \qquad (95)$$

The value of the single-sum term depends on the position of $m$ and $j$. We compute the value of $\frac{\sum_\alpha \langle \boldsymbol{x}_m, \boldsymbol{x}_\alpha, \boldsymbol{x}_j\rangle}{n(m-1)}$ over all six possible cases.

1. $h[m] = 1$. This condition restricts $d < m < d + k/2$. This large case can be divided into three following sub-cases.

   (a) Node $j$ is a child of $m$, i.e. $p[j] = m$. In this case, if $\alpha = j'$ is another child of $m$, then $\langle x_m, x_{j'}, x_j\rangle = 1 - 2q_{m,j',j}$. For all other $\alpha \in P \setminus \{j\}$, observe that the inner product $\langle x_m, x_\alpha, x_j\rangle$ is the product of an even number of relevant input variables, so it is a random variable with mean $1 - \rho$. On the other hand, if $d < \alpha < m$, then $\langle x_m, x_\alpha, x_j\rangle$ is the product of an odd number of relevant input variables, so it is a random variable with mean zero. Therefore, we have

   $$\frac{1}{n(m-1)}\sum_{\alpha}\langle \boldsymbol{x}_m, \boldsymbol{x}_\alpha, \boldsymbol{x}_j\rangle = \frac{1 - 2q_{m,c_1[m],c_2[m]}}{m-1} + \sum_{\alpha \in P \setminus \{j\}}\frac{1 - 2q_{m,\alpha,j}}{m-1}(1 - \rho) + O(\kappa). \qquad (96)$$

   (b) If $h[j] = 0$ but $p[j] \neq m$. In this case, the inner product $\langle x_m, x_\alpha, x_j\rangle$ is never a determined value. Instead, $\langle x_m, x_\alpha, x_j\rangle$ is the product of an even number of relevant input variables whenever $\alpha \in P$, and a variable of mean zero otherwise. Therefore,

   $$\frac{1}{n(m-1)}\sum_{\alpha}\langle \boldsymbol{x}_m, \boldsymbol{x}_\alpha, \boldsymbol{x}_j\rangle = \sum_{\alpha \in P}\frac{1 - 2q_{m,\alpha,j}}{m-1}(1 - \rho) + O(\kappa). \qquad (97)$$

   (c) Finally, suppose $h[j] = 1$. Observe that if $j$ is still an input, then the three-term interaction is a random variable with mean zero; but if $d < \alpha < m$, it is a product of four relevant inputs. Therefore,

   $$\frac{1}{n(m-1)}\sum_{\alpha}\langle \boldsymbol{x}_m, \boldsymbol{x}_\alpha, \boldsymbol{x}_j\rangle = \sum_{\alpha=d+1}^{m-1}\frac{1 - 2q_{m,\alpha,j}}{m-1}(1 - \rho) + O(\kappa). \qquad (98)$$

2. $h[m] > 1$. This condition restricts $d + k/2 < m \leq d + k - 1$. Again, this case can be divided into three following sub-cases.

   (a) Suppose $j \in P$. Then the inner product $\langle x_m, x_\alpha, x_j\rangle$ is always a product of two, four, or six inputs if $\alpha \in P$, or it is a random variable with mean zero otherwise. Therefore,

   $$\frac{1}{n(m-1)}\sum_{\alpha}\langle \boldsymbol{x}_m, \boldsymbol{x}_\alpha, \boldsymbol{x}_j\rangle = \sum_{\alpha \in P}\frac{1 - 2q_{m,\alpha,j}}{m-1}(1 - \rho) + O(\kappa). \qquad (99)$$

   (b) If $p[j] = m$, then clearly $h[j] = h[m] - 1 \geq 1$. If $\alpha = j'$ is another child of $m$, then $\langle x_m, x_{j'}, x_j\rangle = 1 - 2q_{m,j',j}$. On the other hand, if $d < \alpha < m$ and $\alpha \neq j'$, then any inner product is the product of an even number of relevant input variables, so it is a random variable with mean $1 - \rho$. In all other cases, the inner product has mean zero. Hence,

   $$\frac{1}{n(m-1)}\sum_{\alpha}\langle \boldsymbol{x}_m, \boldsymbol{x}_\alpha, \boldsymbol{x}_j\rangle = \frac{1 - 2q_{m,c_1[m],c_2[m]}}{m-1} + \sum_{d < \alpha < m, \alpha \neq j'}\frac{1 - 2q_{m,\alpha,j}}{m-1}(1 - \rho) + O(\kappa). \qquad (100)$$

(c) Now suppose $h[j] > 0$ but $p[j] \neq m$. Then as long as $d < \alpha < m$, any inner product is the product of an even number of relevant input variables, so it is a random variable with mean $1 - \rho$. In all other cases, the inner product has mean zero. Hence,

$$\frac{1}{n(m-1)} \sum_{\alpha} \langle \boldsymbol{x}_m, \boldsymbol{x}_\alpha, \boldsymbol{x}_j \rangle = \sum_{\alpha=d+1}^{m-1} \frac{1 - 2q_{m,\alpha,j}}{m-1}(1-\rho) + O(\kappa). \tag{101}$$

## C.2 The second term

We now evaluate the second term.

$$\frac{1}{n(m-1)} \langle -\mathbf{1}_n + 2\hat{\boldsymbol{z}}_m, 2\hat{\boldsymbol{z}}_m, \boldsymbol{x}_j - \hat{\boldsymbol{z}}_m \rangle \tag{102}$$

$$= \frac{1}{n(m-1)} \langle -\mathbf{1}_n, 2\hat{\boldsymbol{z}}_m, \boldsymbol{x}_j - \hat{\boldsymbol{z}}_m \rangle + \frac{1}{n(m-1)} \langle 2\hat{\boldsymbol{z}}_m, 2d^3\hat{\boldsymbol{z}}_m, \boldsymbol{x}_j - \hat{\boldsymbol{z}}_m \rangle \tag{103}$$

$$= -\frac{2}{n(m-1)} \langle \hat{\boldsymbol{z}}_m, \boldsymbol{x}_j \rangle + \frac{2}{n(m-1)} \langle \hat{\boldsymbol{z}}_m^2 \rangle + \frac{4}{n(m-1)} \langle \hat{\boldsymbol{z}}_m^2, \boldsymbol{x}_j \rangle - \frac{4}{n(m-1)} \langle \hat{\boldsymbol{z}}_m^3 \rangle. \tag{104}$$

We focus on the first and third terms in the final expression. In particular, we compute the following:

$$\frac{1}{n} \langle \hat{\boldsymbol{z}}_m, \boldsymbol{x}_j \rangle = \frac{1}{n(m-1)} \sum_{\alpha} \langle \boldsymbol{x}_\alpha, \boldsymbol{x}_j \rangle \quad \& \quad \frac{1}{n} \langle \hat{\boldsymbol{z}}_m^2, \boldsymbol{x}_j \rangle = \frac{1}{n(m-1)^2} \sum_{\alpha,\beta} \langle \boldsymbol{x}_\alpha, \boldsymbol{x}_\beta, \boldsymbol{x}_j \rangle.$$

For the first two-order term, we divide into two cases.

1. $h[j] = 0$. Clearly, $\langle x_j, x_j \rangle = 1$, and if $\alpha \in P \setminus \{j\}$, the inner product is a product of two relevant bits, so it is a random variable with mean $1 - \rho$. Otherwise, it is a variable with mean zero. Hence,

$$\frac{1}{n} \langle \hat{\boldsymbol{z}}_m, \boldsymbol{x}_j \rangle = \frac{1}{m-1} + \frac{k-1}{m-1}(1-\rho) + O(\kappa). \tag{105}$$

2. $h[j] \geq 1$. Be aware that $x_j$ itself is a product of $2^{h[j]}$ relevant input bits. Again, $\langle x_j, x_j \rangle = 1$. If $h[\alpha] = 0$, then the inner product is a product of an odd number of input bits, so has mean zero. For all other cases, i.e. $d < \alpha < m$ and $\alpha \neq j$, the product is the same as a product of an even number of relevant bits, and has mean $1 - \rho$ without corruption. Therefore,

$$\frac{1}{n} \langle \hat{\boldsymbol{z}}_m, \boldsymbol{x}_j \rangle = \frac{1}{m-1} + \sum_{d < \alpha < m, \alpha \neq j} \frac{1 - 2q_{\alpha,j}}{m-1}(1-\rho) + O(\kappa). \tag{106}$$

For the next three-order term, we again divide into two cases on the height of $j$.

1. $h[j] = 0$. Be aware that $x_j$ itself is a product of $2^{h[j]}$ relevant input bits. Therefore, to transform a product $\langle x_\alpha, x_\beta, x_j \rangle$ to an even multiplication of relevant bits, exactly one of $\alpha$ and $\beta$ must have height at least one, and the other must have height zero. The order can be different, so there are in total $2k(m - d - 1)$ possible combinations. Hence, the total sum in this case is

$$\frac{1}{n} \langle \hat{\boldsymbol{z}}_m^2, \boldsymbol{x}_j \rangle = \frac{2k(1 - 2q_\alpha)}{(m-1)^2}(1-\rho) + O(\kappa). \tag{107}$$

The summands only need to take care of the poisoning rate of the nodes with non-zero height because input bits are not corrupted.

2. $h[j] \geq 1$. In this case, observe that $\langle x_\alpha, x_\beta, x_j \rangle$ has mean $1 - \rho$ if and only if $h[\alpha] = h[\beta] = 0$ or $h[\alpha], h[\beta] \geq 1$. So there are in total $k^2 + (m - d - 1)^2$ possibilities:

$$\frac{1}{n} \langle \hat{\boldsymbol{z}}_m^2, \boldsymbol{x}_j \rangle = \sum_{\alpha,\beta=d+1}^{m-1} \frac{1 - 2q_{\alpha,\beta,j}}{(m-1)^2}(1-\rho) + \sum_{\alpha,\beta \in P} \frac{1 - 2q_j}{(m-1)^2}(1-\rho) + O(\kappa). \tag{108}$$

### C.3 Differences of gradient updates

Given $j \neq j' < m$, we compute the differences of gradient updates for $L$ with respect to $w_{j,m}$ and $w_{j',m}$ as the following:

$$\Delta_{m,j,j'} = \frac{\partial L}{\partial w_{j,m}}(\mathbf{W}) - \frac{\partial L}{\partial w_{j',m}}(\mathbf{W}) \tag{109}$$

$$= -\frac{1}{n(m-1)} \langle \boldsymbol{x}_m, 2\hat{\boldsymbol{z}}_m, \boldsymbol{x}_j - \boldsymbol{x}_{j'} \rangle + \frac{1}{n(m-1)} \langle -\mathbf{1}_n + 2\hat{\boldsymbol{z}}_m, 2d^3\hat{\boldsymbol{z}}_m, \boldsymbol{x}_j - \boldsymbol{x}_{j'} \rangle \tag{110}$$

$$= \left( -\frac{2}{n(m-1)} \langle \boldsymbol{x}_m, 2\hat{\boldsymbol{z}}_m, \boldsymbol{x}_j \rangle \right) - \left( -\frac{2}{n(m-1)} \langle \boldsymbol{x}_m, 2\hat{\boldsymbol{z}}_m, \boldsymbol{x}_{j'} \rangle \right) \tag{111}$$

$$+ \left( \frac{2}{n(m-1)} \langle -\mathbf{1}_n + 2\hat{\boldsymbol{z}}_m, \hat{\boldsymbol{z}}_m, \boldsymbol{x}_j \rangle \right) - \left( \frac{2}{n(m-1)} \langle -\mathbf{1}_n + 2\hat{\boldsymbol{z}}_m, \hat{\boldsymbol{z}}_m, \boldsymbol{x}_{j'} \rangle \right) \tag{112}$$

$$= \left( -\frac{2}{n(m-1)^2} \sum_\alpha \langle \boldsymbol{x}_m, \boldsymbol{x}_\alpha, \boldsymbol{x}_j \rangle \right) - \left( -\frac{2}{n(m-1)^2} \sum_\alpha \langle \boldsymbol{x}_m, \boldsymbol{x}_\alpha, \boldsymbol{x}_{j'} \rangle \right) \tag{113}$$

$$+ \left( -\frac{1}{n(m-1)^2} \sum_\alpha \langle \boldsymbol{x}_\alpha, \boldsymbol{x}_j \rangle \right) - \left( -\frac{1}{n(m-1)^2} \sum_\alpha \langle \boldsymbol{x}_\alpha, \boldsymbol{x}_{j'} \rangle \right) \tag{114}$$

$$+ \left( \frac{1}{n(m-1)^3} \sum_{\alpha,\beta} \langle \boldsymbol{x}_\alpha, \boldsymbol{x}_\beta, \boldsymbol{x}_j \rangle \right) - \left( \frac{1}{n(m-1)^3} \sum_{\alpha,\beta} \langle \boldsymbol{x}_\alpha, \boldsymbol{x}_\beta, \boldsymbol{x}_{j'} \rangle \right) \tag{115}$$

$$+ O(d^{-2-\epsilon/4}). \tag{116}$$

Observe that the first two terms depend on locations of all $\{m, j, j'\}$, while last four terms above do not depend on $m$ but only depend on the locations of $j$ and $j'$. So we compute them separately. In particular, for each choice of $m$, we must first compute the "**correct**" gradient $\partial L / \partial w_{c_1[m],m} = \partial L / \partial w_{c_2[m],m}$, and then compute the "**incorrect**" gradients depending on the location of $j$. Concretely, the steps are the following.

1. Assume $h[m] = 1$, compute $\partial L / \partial w_{c_1[m],m} = \partial L / \partial w_{c_2[m],m}$.
   (a) Compute the gradient $\partial L / \partial w_{j,m}$ if $h[j] = 0$ but $p[j] \neq m$.
   (b) Compute the gradient $\partial L / \partial w_{j',m}$ if $h[j'] > 0$.
   (c) Subtract the correct gradient with the previous two incorrect gradients.
2. Assume $h[m] > 1$, compute $\partial L / \partial w_{c_1[m],m} = \partial L / \partial w_{c_2[m],m}$.
   (a) Compute the gradient $\partial L / \partial w_{j,m}$ if $h[j] = 0$.
   (b) Compute the gradient $\partial L / \partial w_{j',m}$ if $h[j'] > 0$ but $p[j] \neq m$.
   (c) Subtract the correct gradient with the previous two incorrect gradients.

For Step 1, the equation is

$$\left( -\frac{2}{m-1} \right) \times eq. \ (96) + \left( -\frac{2}{m-1} \right) \times eq. \ (105) + \left( \frac{4}{m-1} \right) \times eq. \ (107). \tag{117}$$

For Step 1.(a), the equation is

$$\left( -\frac{2}{m-1} \right) \times eq. \ (97) + \left( -\frac{2}{m-1} \right) \times eq. \ (105) + \left( \frac{4}{m-1} \right) \times eq. \ (107). \tag{118}$$

For Step 1.(b), the equation is

$$\left( -\frac{2}{m-1} \right) \times eq. \ (98) + \left( -\frac{2}{m-1} \right) \times eq. \ (106) + \left( \frac{4}{m-1} \right) \times eq. \ (108). \tag{119}$$

For Step 1.(c), the differences are

$$\left( -\frac{2}{m-1} \right) \times eq. \ (96) - \left( -\frac{2}{m-1} \right) \times eq. \ (97) = -\frac{2\rho(1-2q_m)}{(m-1)^2}; \tag{120}$$

and

$$\left(-\frac{2}{m-1}\right) \times eq.\ (96) - \left(-\frac{2}{m-1}\right) \times eq.\ (98) \tag{121}$$

$$+ \left(-\frac{2}{m-1}\right) \times eq.\ (105) - \left(-\frac{2}{m-1}\right) \times eq.\ (106) \tag{122}$$

$$+ \left(\frac{4}{m-1}\right) \times eq.\ (107) - \left(\frac{4}{m-1}\right) \times eq.\ (108). \tag{123}$$

For Step 2, the equation is

$$\left(-\frac{2}{m-1}\right) \times eq.\ (100) + \left(-\frac{2}{m-1}\right) \times eq.\ (106) + \left(\frac{4}{m-1}\right) \times eq.\ (108). \tag{124}$$

For Step 2.(a), the equation is

$$\left(-\frac{2}{m-1}\right) \times eq.\ (99) + \left(-\frac{2}{m-1}\right) \times eq.\ (105) + \left(\frac{4}{m-1}\right) \times eq.\ (107). \tag{125}$$

For Step 2.(b), the equation is

$$\left(-\frac{2}{m-1}\right) \times eq.\ (101) + \left(-\frac{2}{m-1}\right) \times eq.\ (106) + \left(\frac{4}{m-1}\right) \times eq.\ (108). \tag{126}$$

For Step 2.(c), the differences are

$$\left(-\frac{2}{m-1}\right) \times eq.\ (100) - \left(-\frac{2}{m-1}\right) \times eq.\ (99) \tag{127}$$

$$+ \left(-\frac{2}{m-1}\right) \times eq.\ (106) - \left(-\frac{2}{m-1}\right) \times eq.\ (105) \tag{128}$$

$$+ \left(\frac{4}{m-1}\right) \times eq.\ (108) - \left(\frac{4}{m-1}\right) \times eq.\ (107); \tag{129}$$

and

$$\left(-\frac{2}{m-1}\right) \times eq.\ (100) - \left(-\frac{2}{m-1}\right) \times eq.\ (101) = -\frac{2\rho(1 - 2q_{m,c_1[m],c_2[m]})}{(m-1)^2}. \tag{130}$$

We have computed one gradient difference for each case of $m$, and there is one remaining for each $m$. Observe that, we only need to compute three following expressions:

$$G_{h[m]=1}(m, j, \rho) = -\frac{2}{m-1} \times (eq.\ (96) - eq.\ (98)); \tag{131}$$

$$G_{h[m]>1}(m, j, \rho) = -\frac{2}{m-1} \times (eq.\ (100) - eq.\ (99)); \tag{132}$$

and

$$\left(-\frac{2}{m-1}\right) \times eq.\ (105) - \left(-\frac{2}{m-1}\right) \times eq.\ (106) + \left(\frac{4}{m-1}\right) \times eq.\ (107) - \left(\frac{4}{m-1}\right) \times eq.\ (108). \tag{133}$$

Observe that the remaining gradients can be equivalently expressed as

$$G_{h[m]=1}(m, j, \rho) + eq.\ (133) \quad \& \quad G_{h[m]>1}(m, j, \rho) - eq.\ (133). \tag{134}$$

Using the ingredients from earlier results, for $m \in \{d+1, \ldots, d+k/2\}$ and $d < j < m$, we have

$$G_{h[m]=1}(m, j, \rho) = \frac{-2(1 - 2q_m)}{(m-1)^2} + \frac{-2(k-1)(1 - 2q_m)}{m-1}(1 - \rho)$$
$$- \sum_{\alpha=d+1}^{m-1} \frac{-2(1 - 2q_{m,\alpha,j})}{m-1}(1 - \rho) + O(d^{-2-\epsilon/4}). \tag{135}$$

Similarly, for $m > d + k/2$ and $d < j < m$, we have

$$G_{h[m]>1}(m, j, \rho) = \frac{-2(1 - 2q_{m,c_1[m],c_2[m]})}{(m-1)^2} + \sum_{d < \alpha < m, \alpha \neq j'} \frac{-2(1 - 2q_{m,\alpha,j})}{(m-1)^2}(1 - \rho)$$
$$- \frac{-2k(1 - 2q_{m,\alpha,j})}{(m-1)^2}(1 - \rho) + O(d^{-2-\epsilon/4}). \tag{136}$$

Observe that Equation (105)-(108) can all be factored out by $1 - \rho$, so we may have

$$\text{Equation } (133) = (1 - \rho) \cdot S(m, j) \tag{137}$$

for an expression $S(m, j)$. Using our results of Equation (105)-(108) earlier, we have

$$S(m, j) = -\frac{2(k-1)}{(m-1)^2} + \sum_{d < \alpha < m, \alpha \neq j} \frac{2(1 - 2q_\alpha)}{(m-1)^2} + \sum_{\alpha=d+1}^{m-1} \frac{8k(1 - 2q_\alpha)}{(m-1)^3} - \sum_{\alpha,\beta=d+1}^{m-1} \frac{4(1 - 2q_{\alpha,\beta,j})}{(m-1)^3} - \sum_{\alpha,\beta \in P} \frac{4(1 - 2q_m)}{(m-1)^3}. \tag{138}$$

Finally, for each case of $m$, we conclude the differences between correct and incorrect gradients:

- $h[m] = 1$. Then the differences are

$$\left\{ -\frac{2\rho(1 - 2q_m)}{(m-1)^2} \right\} \quad \& \quad \left\{ G_{h[m]=1}(m, j, \rho) + (1 - \rho)S(m, j), \quad d < j < m \right\}. \tag{139}$$

- $h[m] > 1$. Then the differences are

$$\left\{ -\frac{2\rho(1 - 2q_{m,c_1[m],c_2[m]})}{(m-1)^2} \right\} \quad \& \quad \left\{ G_{h[m]>1}(m, j, \rho) - (1 - \rho)S(m, j), \quad d < j < m \right\}. \tag{140}$$

### C.4 Conditions are sufficient

For any choice of $m$, if the condition in either Equation (7) or (8) (depending on the height of $m$) holds, then the condition is equivalent with the fact: There exists a value $\mu > -2 - \epsilon/4$ such that for any $j < m$, the differences between correct and incorrect gradients satisfy the following

$$\Delta_{m,c_1[m],j}, \Delta_{m,c_2[m],j} < -O(d^\mu). \tag{141}$$

This means the gap between the correct and incorrect gradients is large. If we pick any $\mu' \in (-\mu, 2 + \epsilon/4)$ and choose a learning rate $\eta = \Theta(d^{\mu'})$, then after one gradient update, the difference between weights for children of $m$ and weights for non-children is:

$$\left| \Delta_{m,c_1[m],j}^{(1)} \right|, \left| \Delta_{m,c_2[m],j}^{(1)} \right| \geq O(d^{-\mu+\mu'}) + O(d^{-2-\epsilon/4+\mu'}) = O(d^{-\mu+\mu'}). \tag{142}$$

The last equality holds because, by the range of $\mu'$, we must have $-\mu + \mu' > 0$ and $-2 - \epsilon/4 + \mu' < 0$; so the second quantity is dominated by the first one.

The conditions imply that the incorrect weights are smaller than correct weights, so applying the softmax attention score function, for $j \neq c_1[m], c_2[m]$ we have

$$\sigma_j(\boldsymbol{w}_m^{(1)}) \leq \exp\left( -\left| \Delta_{m,c_1[m],j}^{(1)} \right| \right) \leq \exp\left( -\Theta\left( d^{-\mu+\mu'} \right) \right). \tag{143}$$

Softmax scores must sum to 1, we must have

$$\sigma_{c_1[m]}(\boldsymbol{w}_m^{(1)}) + \sigma_{c_2[m]}(\boldsymbol{w}_m^{(1)}) \geq 1 - \exp\left( -\Theta\left( d^{-\mu+\mu'} \right) \right). \tag{144}$$

Moreover, we observe that in this case, the correct attention scores $\sigma_{c_1[m]}(\boldsymbol{w}_m^{(1)})$ and $\sigma_{c_2[m]}(\boldsymbol{w}_m^{(1)})$ are close enough:

$$\frac{\sigma_{c_1[m]}(\boldsymbol{w}_m^{(1)})}{\sigma_{c_2[m]}(\boldsymbol{w}_m^{(1)})} = \exp\left( w_{c_1[m],m}^{(1)} - w_{c_2[m],m}^{(1)} \right) \leq \exp\left( O\left( d^{-2-\epsilon/4+\mu'} \right) \right) \leq 1 + O\left( d^{-2-\epsilon/4+\mu'} \right), \tag{145}$$

where the last inequality holds because $e^t \leq 1 + O(t)$ for small $t > 0$. By symmetry, we have the same upper bound for $\sigma_{c_1[m]}(\boldsymbol{w}_m^{(1)})/\sigma_{c_2[m]}(\boldsymbol{w}_m^{(1)})$. As a result, we have

$$\frac{1}{2} - O\left(d^{-2-\epsilon/4+\mu'}\right) \leq \sigma_{c_1[m]}(\boldsymbol{w}_m^{(1)}), \sigma_{c_2[m]}(\boldsymbol{w}_m^{(1)}) \leq \frac{1}{2} + O\left(d^{-2-\epsilon/4+\mu'}\right). \tag{146}$$

The equation above implies that for each step $d < m \leq d + k - 1$, the attention layer at step $m$ almost computes the average of two children nodes, and all information from other non-children nodes are dominated and essentially vanish as $d$ becomes large.

We now show that using the attention scores, every prediction step, including the final output, has a vanishing loss. Let $\boldsymbol{x}$ be a $d$-dimensional binary input vector. For every $d < m \leq d + k - 1$, let $\hat{z}_m^{(1)}$ be the empirical output of the attention layer, then $\phi(\hat{z}_m^{(1)})$ is the empirical prediction, and the prediction loss for step $m$ is

$$\epsilon_m = \left| \phi\left(\hat{z}_m^{(1)}\right) - \phi\left(\frac{x_{c_1[m]} + x_{c_2[m]}}{2}\right) \right| \tag{147}$$

$$\leq 2 \times \left| \hat{z}_m^{(1)} - \frac{x_{c_1[m]} + x_{c_2[m]}}{2} \right| \tag{148}$$

$$\leq 2 \times \left| \hat{z}_m^{(1)} - \frac{\hat{x}_{c_1[m]} + \hat{x}_{c_2[m]}}{2} \right| + 2 \times \left| \frac{x_{c_1[m]} + x_{c_2[m]}}{2} - \frac{\hat{x}_{c_1[m]} + \hat{x}_{c_2[m]}}{2} \right| \tag{149}$$

$$= 2d \exp\left(-\Theta\left(d^{-\mu+\mu'}\right)\right) + 2 \times \left| \sigma_{c_1[m]}(\boldsymbol{w}_m^{(1)}) - \frac{1}{2} \right| + 2 \times \left| \sigma_{c_2[m]}(\boldsymbol{w}_m^{(1)}) - \frac{1}{2} \right| + 2\epsilon_{m-1} \tag{150}$$

$$= O(d^{-2-\epsilon/4+\mu'}) + O(\epsilon_{m-1}). \tag{151}$$

If $m = d + 1$, then $\epsilon_{m-1} = \epsilon_d = 0$ because the $d$-th value is still an input. Therefore, this upper bound holds for every $d < m \leq d + k - 1$ and take $m = d + k - 1$ so that $x_m = y$, we conclude that $|\hat{y} - y| = |\hat{x}_{d+k-1} - x_{d+k-1}| = O(d^{-2-\epsilon/4+\mu'})$.

## C.5 Conditions are necessary

Now suppose there is at least one $m$ such that the condition in Equation (7) or (8) (depending on the height of $m$) does not hold. This implies that, for this $m$, there exists at least one non-child node $j$ such that $j < m$ and the gap between this incorrect gradient $\partial L/\partial w_{m,j}$ and the correct gradients $\partial L/\partial w_{m,c_1[m]}$, $\partial L/\partial w_{m,c_2[m]}$ is too small to be distinguished. Precisely, there exists a number $\delta \leq -2 - \epsilon/4$ such that

$$\left| \Delta_{m,c_1[m],j}^{(1)} \right|, \left| \Delta_{m,c_2[m],j}^{(1)} \right| \leq O(d^\delta) + O(d^{-2-\epsilon/4}) = O(d^{-2-\epsilon/4}) = \left| \Delta_{m,c_1[m],c_2[m]}^{(1)} \right|. \tag{152}$$

Using the same analysis in the proof of sufficiency, the small gaps implies that the attention scores $\sigma_{c_1[m]}(\boldsymbol{w}_m^{(1)}), \sigma_{c_2[m]}(\boldsymbol{w}_m^{(1)}), \sigma_j(\boldsymbol{w}_m^{(1)})$ are close. Consider the optimal scenario under this case, that the condition in Equation (7) or (8) (depending on the height of $m$) holds for any other $j' \neq j$, then we have $\sigma_{c_1[m]}(\boldsymbol{w}_m^{(1)}) + \sigma_{c_2[m]}(\boldsymbol{w}_m^{(1)}) + \sigma_j(\boldsymbol{w}_m^{(1)}) = 1 - e^{-\Theta(d^\nu)}$ for some $\nu > 0$ and for any $a, b \in \{c_1[m], c_2[m], j\}$, we have

$$\frac{\sigma_a(\boldsymbol{w}_m^{(1)})}{\sigma_b(\boldsymbol{w}_m^{(1)})} \leq \exp\left(O\left(d^{-2-\epsilon/4}\right)\right) \leq 1 + O\left(d^{-2-\epsilon/4}\right). \tag{153}$$

Equivalently,

$$\frac{1}{3} - O\left(d^{-2-\epsilon/4}\right) \leq \sigma_{c_1[m]}(\boldsymbol{w}_m^{(1)}), \sigma_{c_2[m]}(\boldsymbol{w}_m^{(1)}), \sigma_j(\boldsymbol{w}_m^{(1)}) \leq \frac{1}{3} + O\left(d^{-2-\epsilon/4}\right) \tag{154}$$

For a sufficiently large $d$, i.e. as $d \to \infty$, we may regard the predictor at step $m$ is exactly the function $\hat{x}_m = \phi\left(\frac{1}{3}x_{c_1[m]} + \frac{1}{3}x_{c_2[m]} + \frac{1}{3}x_j\right)$, where the ground truth must still be $x_m = \phi\left(\frac{1}{2}x_{c_1[m]} + \frac{1}{2}x_{c_2[m]}\right)$. If the $d$-dimensional inputs are uniformly generated, then with probability exactly 0.5, the sample satisfies the property that $x_{c_1[m]} = -x_{c_2[m]}$, so the true prediction for step $m$ is $x_m = \phi(0) = -1$. On the other hand, the

empirical prediction is $\hat{x}_m = \phi(x_j/3)$ and therefore fixed as $\phi(1/3) = \phi(-1/3) = -1/3$. Therefore, the the error for prediction at stepm $m$ is lower bounded as the following:

$$\mathbb{E}_{\boldsymbol{x} \sim \mathrm{Uniform}(\{\pm1\}^d)} |\hat{x}_m - x_m| \geq \frac{1}{2} \times \left| -1 + \frac{1}{3} \right| = \frac{1}{3} = \Omega(1). \tag{155}$$

We conclude this proof by showing that, if one step $m$ for some $d < m \leq d + k - 1$ has a non-negligible loss, then the loss for the final prediction also has a non-negligible loss, i.e. $|y - \hat{y}| = |x_{d+k-1} - \hat{x}_{d+k-1}| = \Omega(1)$.

We first show that such an error of node $m$ causes a non-negligible damage on its parent node $p[m]$, i.e. $|x_{p[m]} - \hat{x}_{p[m]}| = \Omega(1)$. Denote $m'$ as the unique sibling of $m$, i.e. $p[m] = p[m']$. Then clearly $x_{p[m]} = x_m x_{m'}$. Hence, the following inequality satisfies; we abuse the notation by using $\mathbb{E}$ as the uniform generation.

$$\mathbb{E}_{\boldsymbol{x} \sim \mathrm{Uniform}(\{\pm1\}^d)} |x_{p[m]} - \hat{x}_{p[m]}| = \mathbb{E}|x_m x_{m'} - \hat{x}_m \hat{x}_{m'}| \tag{156}$$

$$\overset{d \to \infty}{=} \mathbb{E}|x_m x_{m'} - x_m \hat{x}_{m'}| \tag{157}$$

$$= \mathbb{E}\left[ |x_m| \cdot |x_{m'} - \hat{x}_{m'}| \right] \tag{158}$$

$$= \mathbb{E}|x_{m'} - \hat{x}_{m'}| = \Omega(1). \tag{159}$$

The inequality holds for every $m$ regardless of its location, and inductively, this non-negligible error propagates to higher ancestors of $m$, and ultimately to the root of the tree.

Recall that this is the "best" scenario when the condition in Equation (7) or (8) fails for $m$, i.e. only one non-child node $j$ has a prohibitively high attention score. If more non-children nodes fail, say $f$ of them in the set $F \subseteq [m-1]$, then $\hat{z}_m = \sum_{f \in F} x_f/(f-1)$. By Hoeffding's inequality, we have

$$\mathbb{P}\left( \phi\left( \frac{\sum_{f \in F} x_f}{f - 1} \right) < 0 \right) = \mathbb{P}\left( \frac{\sum_{f \in F} x_f}{f - 1} < \frac{1}{2} \right) \geq 1 - e^{-\Omega(f)}. \tag{160}$$

Nevertheless, the true prediction is still uniform, so

$$\mathbb{E}|x_m - \hat{x}_m| \geq \frac{1}{2}\left( 1 - e^{-\Omega(f)} \right) = \frac{1}{2} - o(1) = \Omega(1). \tag{161}$$

Using the same argument for the root prediction for the case above, the final prediction also suffers an error of $\Omega(1)$, as desired.

## D   Additional experiment details

The experiments were ran for five times with $d = 128$ and $k = 64$. The mean and variance values for each case is illustrated in Figure 5. For each grid, the mean and variance are computed by the following standard formulas:

$$\mathrm{Mean}(\mu) = \frac{\sum_{i=1}^{n} x_i}{n} \quad \& \quad \mathrm{Variance}(\sigma^2) = \frac{\sum_{i=1}^{n} (x_i - \mu)^2}{n}.$$

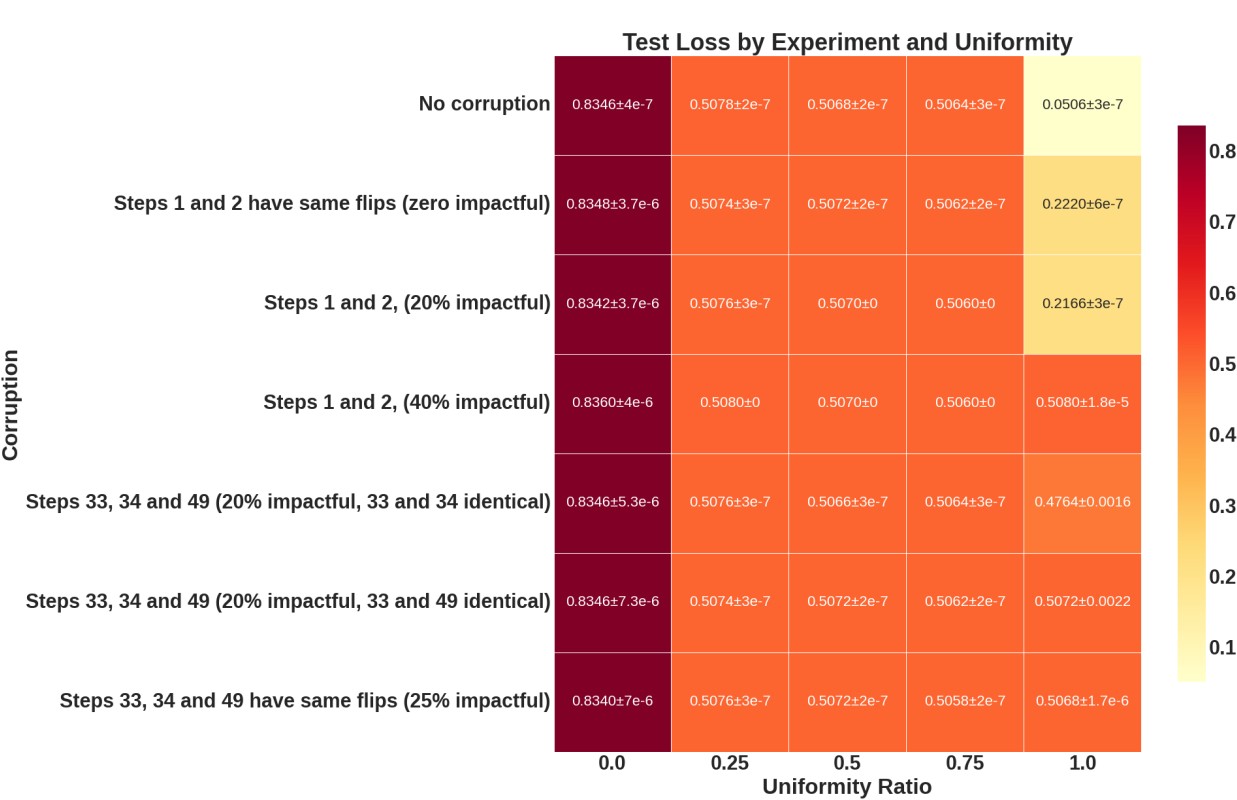

Figure 6: Mean and variance of the test losses after repetitions

