# OpenReview forum: "Data Shifts Hurt Chain-of-Thought: A Case Study on Parity Learning"
_TMLR — Decision pending for TMLR_

### Review · Reviewer_BfhY · 2026-03-31

**Summary Of Contributions:**

The paper studies the impact of training data and data shift on chain of thought (CoT) for the toy task of learning bit subset parity. An imbalanced version of subset parity is introduced, where the data is constrained to be equal on the subset with a fixed probability $\rho$ (information leakage). It is shown that the case $\rho=\Theta(1)$ is solved by a single-layer transformer with one gradient update. Under label noise and information leakage, necessary and sufficient conditions for CoT learning are established, showing that distribution shift always hurts training.

**Strengths**

* The paper extends existing analyses on learning multi-step problems with chain of thought, which generally only consider uniform data distribution, to explicitly focus on data shift and label noise.
* Necessary and sufficient conditions for successful learning are established rigorously, showing the detriments of distribution shift even with information leakage.
* The paper is overall clearly written and easy to follow. Synthetic experiments support the theoretical analysis.

**Weaknesses**

* The described transformer model, training loss, and overall proof techniques all seem very similar to the previous work by Kim and Suzuki (2025), with the main difference being the data distribution. This limits the novelty of the work.
* While it is nice that Theorem 4.2 yields an exact equivalence condition, besides the two extreme cases $\rho=0,1$, it is unclear how to interpret the inequalities Eq.(7),(8) beyond the qualitative explanation that low $\rho$ leads to higher $B_m$. It is rather intuitive that more distribution shift (here essentially defined as label noise) hurts training; is there a sense as in how much is "tolerable"?
* The equivalence statement and resulting lower bound (Corollary 4.3) only holds for the described transformer model, which uses specific non-standard constructions, e.g., the embedding dimension depends on the sample size $n$ and the feedforward layer is fixed depending on problem size $d$. It is unlikely that this failure translates to actual learning-theoretic hardness in a general sense. In constrast, the lower bound in Kim and Suzuki (2025) are shown to hold for all first-order algorithms, making the separation more convincing.

**Audience:**

Yes

**Audience Explanation:**

The parity model, while simplified, has been demonstrated to provide quantitative understanding into optimization dynamics of attention-based models and information-theoretic difficulty of learning classes of functions. The paper specifically studies the new setting of data shift and imbalanced data, which has not been studied before. Understanding the effect of the data distribution on post-training is an active area with great potential practical impact.

**Claims And Evidence:**

Yes

**Claims Explanation:**

The proofs provided in the appendix seem rigorous and correct. The computations are generally similar to that of previous work.

**Requested Changes:**

* The discussion of the main result (Theorem 4,2) should be made more quantitative. From just the inequalities Eq.(7),(8) and definitions $G(m,j,\rho)$ in Eq.(4),(6), it is unclear what the quantifiable takeaways are beyond that data shift hurts training.

* Due to the similarities regarding the model, setup, proof techniques, etc., a separate paragraph or subsection detailing the similarities and differences with Kim and Suzuki (2024) should be included, rather than simply stating they only consider uniform data.

* The proposed data poisoning model should be related/compared to the wide body of existing literature on label noise for classification.

---

> ### Author Response · Authors · 2026-04-19
> **Response to Reviewer BfhY**
>
> We thank the reviewer for the positive assessment and for the constructive suggestions to strengthen the paper. We address each concern below.
>
> > The described transformer model, training loss, and overall proof techniques all seem very similar to the previous work by Kim and Suzuki (2025), with the main difference being the data distribution. This limits the novelty of the work.
>
> We appreciate this concern and will add a dedicated subsection to elaborate the similarities and differences. To clarify the relationship:
>
> Similarities: We study the same one-layer transformer architecture and adopt the same binary-tree CoT decomposition. Several proof ideas (softmax attention score derivatives, positional encoding structure) are shared.
>
> Key differences:
> 1. Kim & Suzuki (2025) study only the uniform distribution (ρ=1) with perfectly correct training data. Our work introduces two new dimensions: (a) distribution shift, and (b) data poisoning. We also characterize their joint impact.
>
> 2. Despite the similar high-level idea, our technical analysis is substantially different because the gradient computations under non-uniform distributions produce terms with non-zero means. This factor does not appear in the uniform case, and demands more careful analyses and computations. Moreover, the poisoning structure, together with the sets $U_{a,b,c}$ and quantities $q_{a,b,c}$, requires a new combinatorial analysis of how corruptions interact across the tree. One example is the analysis in Equations 96-101 (to compute the term in Equation 95) on Pages 21 and 22: Due to non-zero means, the computations needed much more care than the zero-mean case.
>
> In summary, we would claim that despite similarities on the surface, the new dimensions make qualitative distinctions compared to earlier works in the same topic.
>
> > While it is nice that Theorem 4.2 yields an exact equivalence condition, besides the two extreme cases $\rho = 0,1$,
> , it is unclear how to interpret the inequalities Eq.(7),(8) beyond the qualitative explanation that low $\rho$
>  leads to higher $B_m$
>
> We appreciate this important question.
>
> For intermediate values of $\rho$, the quantity $B_m$ in Equations (7)-(8) measures the **worst-case gradient confusion** at the intermediate CoT step $m$. Specifically, it measures the smallest gap between the gradient update for the true children and that for the “**best-mimicking**” non-child node. A more negative $B_m$ (with a larger absolute value) means better separation and therefore easier learning; a $B_m$ with smaller absolute value means the model cannot distinguish correct children nodes from incorrect non-children nodes.

---

> ### Author Response · Authors · 2026-04-19
> **Response to Reviewer BfhY (Continued)**
>
> > It is rather intuitive that more distribution shift (here essentially defined as label noise) hurts training; is there a sense as in how much is "tolerable"?
>
> We appreciate this important question.
>
> **The impact of distribution shift is counterintuitive, not intuitive**. The review interprets the distribution shift (quantified by the value of $\rho$) as “essentially label noise” and states it is "rather intuitive" that it hurts training. We respectfully clarify that the analogy is not true, and this characterization of distribution shift is the opposite as our results intended to convey.
>
> In one sentence, the parameter $\rho$ controls how much structural information is leaked about the positions of the relevant bits. Concretely, when $\rho < 1$, the relevant bits are correlated among themselves: They have an identical value (all +1 or all −1) with probability $1-\rho$.
>
> Consider this example: Suppose we are detectives, and the mission is to target a group of criminals (relevant bits) on a very crowded street (all bits).
> * When $\rho=1$, the criminals will choose their outfits randomly (all bits, relevant or irrelevant, are randomly sampled) so they will look indistinguishable from other citizens on the street. In this case, it is difficult to target the criminals.
> * However, when $\rho<1$, then with a non-zero probability $1-\rho$, all criminals will choose an identical outfit (all relevant bits will have values +1 or -1), and this choice would make the entire group way more noticeable.
>
> Therefore, the standard intuition would be that CoT should also benefit from this extra information, or at worst be unaffected. Our central finding is that this intuition is wrong: the same information leakage that enables direct prediction always **hurts** CoT training (Theorem 4.2), and at maximum leakage ($\rho=0$), CoT fails completely (Corollary 4.3), even though this case is, by intuition, the easiest case.
>
> The reason for this counter-intuition is that: When relevant bits are correlated, the gradient updates during CoT steps extract **nearly identical** signals from correct children and incorrect non-children, so the children and non-children are **indistinguishable**. This creates a scenario where CoT is strictly worse than direct prediction not because the problem is hard, but because the extra information disrupts the CoT decomposition's ability to identify the correct tree structure.
>
> We acknowledge that this point was insufficiently emphasized in the draft and we will add more explanations in the revision.
>
> > The equivalence statement and resulting lower bound (Corollary 4.3) only holds for the described transformer model, which uses specific non-standard constructions, e.g., the embedding dimension depends on the sample size $n$
>  and the feedforward layer is fixed depending on problem size $d$
> . It is unlikely that this failure translates to actual learning-theoretic hardness in a general sense. In constrast, the lower bound in Kim and Suzuki (2025) are shown to hold for all first-order algorithms, making the separation more convincing.
>
> We acknowledge that our architecture-specific constructions mean the lower bound is specific to this model class rather than a general learning-theoretic hardness result. We will discuss this distinction explicitly. That said, architecture-specific analysis is the norm in the theoretical transformer literature (Zhang et al., 2024; Huang et al., 2024; Kim & Suzuki, 2024, 2025). General hardness results typically require fundamentally different techniques (e.g., statistical query lower bounds). Nevertheless, we view establishing architecture-agnostic lower bounds under data shifts as an important direction for future work, and will state this explicitly.
>
> > The discussion of the main result (Theorem 4,2) should be made more quantitative. From just the inequalities Eq.(7),(8) and definitions $G(m,j,\rho)$
>  in Eq.(4),(6), it is unclear what the quantifiable takeaways are beyond that data shift hurts training.
>
> Thanks for this suggestion. We will add more quantitative discussions in our revision.
>
> > Due to the similarities regarding the model, setup, proof techniques, etc., a separate paragraph or subsection detailing the similarities and differences with Kim and Suzuki (2024) should be included, rather than simply stating they only consider uniform data.
>
> Thanks for this suggestion. We agree and will add a separate subsection explaining this.
>
> > The proposed data poisoning model should be related/compared to the wide body of existing literature on label noise for classification.
>
> Thanks for this suggestion. We will add a discussion to explain the relationships between our data poisoning model to the broader literature on label noise.

---

### Review · Reviewer_L7ve · 2026-04-01

**Summary Of Contributions:**

This paper studies the effect of training-time data shifts on chain-of-thought learning for the generalised k-parity problem. The setup considers two deviations from the ideal setting used in prior theories: distribution shift and poisoning/corruption of the intermediate CoT supervision. The paper first shows that for imbalanced parity distributions, a one-layer transformer can solve the task efficiently without CoT. It then analyses a specific known CoT decomposition for parity and gives a necessary-and-sufficient condition for successful learning under joint distribution shift and poisoning.
Within this setup, distribution shift always hurts CoT training, and sufficiently severe poisoning at any intermediate step causes failure, with tolerance scales as O(1/k). The paper also includes small-scale synthetic experiments intended to support the theory.

Strengths:
- The paper has a well-defined theoretical setup - it formulates a precise learning problem (generalised parity_ with explicit assumptions on data distribution, supervision and model class.
- The finding that this CoT decomposition can be strictly worse than direct prediction on imbalanced parity is genuinely interesting and counterintuitive.

Weakness
- The scope is narrow: the results apply only to a specific CoT decomposition of k-parity using a specific one-layer transformer architecture with a specific feedforward function, all-zero initialisation, and a single gradient step. While the paper speaks as though the conclusions apply to CoT more generally, but in fact the degree to which these insights from k-parity problem generalise is unclear. The central message - that data shifts hurt CoT, is not well supported beyond this narrow setting.
- Parity is also not a particularly representative task for understanding why CoT helps in modern ML systems. In parity, the "reasoning chain" is simply a rigid sequence of running XOR operations. There is no semantic abstraction, no branching structure, and multiple valid reasoning paths, etc. The CoT here is best thought of as a hard-coded sequential algorithm. As a result, showing that such a pipeline is brittle under noise is not very surprising. This matters because the practical case for CoT in LLMs comes from tasks where intermediate steps are useful not only because they serialise computation, but because they expose latent structure, support abstraction, allow model to organise problem into subproblems, etc. The parity setting lacks these properties.
- The experiments are limited and do not substantially broaden the scope of the theory. They support the existence of the phenomenon in the toy setting, but they do not do much to argue for broader relevance.

**Additional Comments:**

I want to emphasise that I do not think the paper is uninteresting. In fact, there is a legitimate and subtle point made here: once one imposes a rigid chain of intermediate targets, one may create a supervision structure that is more fragile than direct outcome prediction. This is a nice insight.
The main concern is not with this particular result itself, but with the jump from the result to a broad interpretation about CoT.

**Audience:**

Yes

**Audience Explanation:**

This is a relevant TMLR paper because it concerns the theoretical understanding of transformers, CoT, and supervision under imperfect data conditions. The question is important, and even a narrow negative result can be valuable if framed properly.

**Broader Impact Concerns:**

I do not see major direct broader-impact concerns.

**Claims And Evidence:**

No

**Claims Explanation:**

The paper provides solid evidence for its formal claims within the exact generalised parity setup it studies. Though the experiment is still quite narrow - with specific choice of problem, model architecture and feedforward function.
However, the broader message, which is the key claim of the paper "data shifts hurt CoT" is not supported.

**Requested Changes:**

- Narrow the paper's framing: the title, abstract and conclusion should be calibrated to the actual scope of the results. The paper studies one specific COT decomposition for generalise parity under a very particular training setup. The claims should consistently reflect that. (Critical)
- The paper should openly discuss why parity is or is not a n appropriate model for CoT. Right now, the reason of choosing this problem is because it could be solved by a single-layer transformer. It implicitly treats it as a natural proxy for reasoning, which is not obvious. The authors should explain what aspects of real CoT they believe this setup captures, and what are the limitations. (Critical)
- The paper currently risks attributing the failure to CoT as a concept, when it may instead be caused by the brittleness of this specific decomposition. This should be discussed much more explicitly. (beneficial)
- Discussion and demonstrative experiments of alternative CoT-like supervision paradigms. (beneficial)

---

> ### Author Response · Authors · 2026-04-19
> **Responses to Reviewer L7ve**
>
> We thank the reviewer for the thoughtful feedback. We especially appreciate the remark that "once one imposes a rigid chain of intermediate targets, one may create a supervision structure that is more fragile than direct outcome prediction", though we may address a more precise calibration: This is indeed the central insight regarding the distribution shift impact we hope to convey, via Corollary 4.3. We address each concern below.
>
> >The scope is narrow: the results apply only to a specific CoT decomposition of k-parity using a specific one-layer transformer architecture with a specific feedforward function, all-zero initialisation, and a single gradient step. While the paper speaks as though the conclusions apply to CoT more generally, but in fact the degree to which these insights from k-parity problem generalise is unclear. The central message - that data shifts hurt CoT, is not well supported beyond this narrow setting.
>
> >Narrow the paper's framing: the title, abstract and conclusion should be calibrated to the actual scope of the results. The paper studies one specific COT decomposition for generalise parity under a very particular training setup. The claims should consistently reflect that. (Critical)
>
> We agree and will revise the title, abstract, and conclusions to accurately reflect the scope. We will clarify that our results apply to a specific CoT decomposition of $k$-parity under a well-defined training setup, and our work’s broader implications for CoT are suggestive rather than conclusive. We will use a new title **Data Shifts Hurt Chain-of-Thought: A Case Study on Parity Learning** to emphasize our scope on analyzing the parity problem.
>
> > Parity is also not a particularly representative task for understanding why CoT helps in modern ML systems. In parity, the "reasoning chain" is simply a rigid sequence of running XOR operations. There is no semantic abstraction, no branching structure, and multiple valid reasoning paths, etc. The CoT here is best thought of as a hard-coded sequential algorithm. As a result, showing that such a pipeline is brittle under noise is not very surprising. This matters because the practical case for CoT in LLMs comes from tasks where intermediate steps are useful not only because they serialise computation, but because they expose latent structure, support abstraction, allow model to organise problem into subproblems, etc. The parity setting lacks these properties.
>
> > The paper should openly discuss why parity is or is not a n appropriate model for CoT. Right now, the reason of choosing this problem is because it could be solved by a single-layer transformer. It implicitly treats it as a natural proxy for reasoning, which is not obvious. The authors should explain what aspects of real CoT they believe this setup captures, and what are the limitations. (Critical)
>
> This is an important point. We will add a dedicated discussion addressing this. Our explanations are the following:
>
> What parity **DOES** capture: The core computational structure of CoT is autoregressive generation of intermediate tokens where each step depends on previous steps, trained with process supervision (teacher forcing). This is indeed captured by the $k$-parity problem: The binary-tree decomposition is a clean instance of breaking a hard problem (uniform $k$-parity) into easier subproblems (uniform 2-parity) solved sequentially. Our fragility insights are derived from this sequential dependency structure, while the XOR properties are helpful to the mathematical analyses only. Specifically, the $O(1/k)$ poisoning tolerance arises because errors at any intermediate step propagate through the chain, and this phenomenon holds whenever CoT steps are causally dependent.
>
> What parity **DOES NOT** capture: The reviewer is correct that parity lacks the central components of natural language-based CoT, including semantic abstraction, branching structure, and multiple valid reasoning paths. Natural language-based CoT benefits from (1) redundancy (multiple paths to a correct answer), and (2) intermediate steps that expose latent structure in addition to sequences of computation. Our rigid decomposition has no such redundancy, which likely makes it more fragile than natural language-based CoT. Nevertheless, we view this as a strength of our result as a lower bound on vulnerability: If even a clean, well-structured decomposition (tree-decomposition of $k$-parity) is brittle to this extent, it raises questions about whether similar vulnerabilities exist in more complex settings.
>
> We will add this discussion explicitly in the revised manuscript.

---

> ### Author Response · Authors · 2026-04-19
> **Responses to Reviewer L7ve (Continued)**
>
> > The paper currently risks attributing the failure to CoT as a concept, when it may instead be caused by the brittleness of this specific decomposition. This should be discussed much more explicitly. (beneficial)
>
> We agree this distinction is important. We will add discussions on:
> * The identified failure is specific to the rigid binary-tree decomposition from Kim & Suzuki (2025). Alternative decompositions, such as a flatter tree or one with redundant steps, might interact with data shifts differently.
> * The paradox at $\rho=0$ (Corollary 4.3) is a property of how this specific decomposition's gradient structure interacts with the imbalanced distribution, and it is not necessarily a universal property of all CoT approaches.
> * That said, our results on data poisoning revealed that supervision on a sequential chain means any single corrupted step can mislead the entire computation. This is the **underlying reason** for this fragility, and we believe this reason is structural and not unique to our decomposition and the parity problem.
>
> > The experiments are limited and do not substantially broaden the scope of the theory. They support the existence of the phenomenon in the toy setting, but they do not do much to argue for broader relevance.
>
> > Discussion and demonstrative experiments of alternative CoT-like supervision paradigms. (beneficial)
>
> Thanks for the suggestion. There have been some experimental results on CoT harmness on various settings, yet there has not been sufficient theoretical analyses on this phenomenon. Such experimental results include:
> * Lu, et al. 2025. Does Chain-of-Thought Reasoning Really Reduce Harmfulness from Jailbreaking?
> * Liu, et al. 2024. Mind Your Step (by Step): Chain-of-Thought can Reduce Performance on Tasks where Thinking Makes Humans Worse
> * Liu, et al. 2026. Structural Anchors and Reasoning Fragility: Understanding CoT Robustness in LLM4Code
> * Feng, et al. 2025. What Characterizes Effective Reasoning? Revisiting Length, Review, and Structure of CoT
>
> We will add a more detailed section in Related Works discussing the harm caused by CoT in broader settings.

---

### Review · Reviewer_REZ6 · 2026-04-05

**Summary Of Contributions:**

This work is a focused study of the k-parity problem and the properties of the transformer architecture with a particular output token structure. It makes several theoretical claims regarding distribution shifts (changing a particular parameter of the k-parity problem) and data poisoning (adding noise to the training data).

Strengths:
* Theoretical analysis of the properties of the one-layer transformer architecture with a gradient descent optimizer on a subset of the k-parity problem.
* Theoretical analysis of the effects of distribution shift and data poisoning on the performance of the one-layer transformer architecture for the k-parity problem.
* Early experimental results related to the theoretical analysis.

Weaknesses:
* The scope and terminology used throughout the paper (LLM, CoT) do not reflect the actual content of the paper. This work is focused only on the k-parity problem, a one-layer transformer model (not an LLM), and a specific structure of predictions (not general CoT).
* The relevance of its findings is limited to a narrow audience (likely only those prior works cited in the paper related to the k-parity problem and transformer architectures).
* Derivations are dense with non-obvious expressions that are given with little explanation. The reviewer has checked the majority of the derivations in the main paper to the best of their ability, but not in the appendices.

**Audience:**

Yes

**Audience Explanation:**

Yes, but only to a very limited audience. The results in this paper would likely only be of interest to those familiar with the immediately related prior works on k-parity and transformer architectures, e.g., Kim and Suzuki, 2024, Huang et al., 2024.

**Claims And Evidence:**

No

**Claims Explanation:**

Claims for the theoretical results are supported by the proofs given in the paper.

The evidence for the paper's claims regarding Chain-of-Thought (CoT) prompting (which the paper appears to referenece via Wei et al., 2022) is very weak. This paper does not study Large Language Models (LLMs); it studies a one-layer transformer that does not output language, where CoT is a structured prediction sequence. There are no notations of pre-training, prompting, etc. It is a vast leap of faith, for which the paper provides no evidence, to believe that any of the claims regarding distribution shift or data poisoning have relevance to LLMs with CoT prompting.

This is not to say the paper is not relevant or its claims are not supported, but rather that its claims with regard to LLMs and CoT as most researchers in the field understand them, are not supported. A proper reframing of this work could help resolve this.

**Requested Changes:**

[major] Reduce the scope and claims of this paper to k-parity problem and (one-layer) transformer architectures. The relationship to LLMs and CoT prompting is tenuous and not supported by the rest of the paper at all. This should be moved up-front and ideally in the title.

[major] In Sec. 3.2 "Task decomposition as CoT" does not explain why this is chain of thought? You mention "The chain of thought protocol decomposes the k-parity problem as a sequence of 2-parity problems" but why? What do you mean by the "chain of thought protocol"? Certainly not in the Wei et al. sense? There are no thoughts possible here? No pre-training? In the reviewer's opinion there is no clear link between the decomposition and the commonly understood CoT setting, this should be made clear throughout the paper (see above concern).

[minor] Sec 3.1. Specification of the k-parity problem could be more precise, e.g., what is x_j, where it first appears? what exactly is the task in the current notation?

[minor] Sec 3.2. There are several assumptions here, e.g., "we assume k=2^v for simplicity", are they without loss of generality? Please comment.

[minor] Sec. 4.2 is very difficult to follow, especially when it gets to "Examples for the cases.", the expressions for the gradients are non-obvious, but shown without any explanation.

[minor] Move experimental results into the body of the paper, for space, suggest moving more of the derivations in 4.2 to appendix.

[nit] Sec. 3 notation "any integer n" -> "any positive integer n > 0"?

---

> ### Author Response · Authors · 2026-04-19
> **Responses to Reviewer REZ6**
>
> We thank the reviewer for the careful reading and checking on the derivations. We especially appreciate recognition of the theoretical contribution to the $k$-parity problem and our analysis of distribution shift and poisoning for the transformer architecture. We also appreciate the reviewer’s constructive suggestions. We address each concern below.
>
> > [major] Reduce the scope and claims of this paper to k-parity problem and (one-layer) transformer architectures. The relationship to LLMs and CoT prompting is tenuous and not supported by the rest of the paper at all. This should be moved up-front and ideally in the title. We will use a new title **Data Shifts Hurt Chain-of-Thought: A Case Study on Parity Learning** to emphasize our scope on analyzing the parity problem.
>
> > [major] In Sec. 3.2 "Task decomposition as CoT" does not explain why this is chain of thought? You mention "The chain of thought protocol decomposes the k-parity problem as a sequence of 2-parity problems" but why? What do you mean by the "chain of thought protocol"? Certainly not in the Wei et al. sense? There are no thoughts possible here? No pre-training? In the reviewer's opinion there is no clear link between the decomposition and the commonly understood CoT setting, this should be made clear throughout the paper (see above concern).
>
> > The scope and terminology used throughout the paper (LLM, CoT) do not reflect the actual content of the paper. This work is focused only on the k-parity problem, a one-layer transformer model (not an LLM), and a specific structure of predictions (not general CoT).
>
> We appreciate this feedback and agree that our framing could be more precise. We want to clarify that our notion of "CoT" follows the specific technical definition established in recent theoretical transformer literature, particularly Wies et al. (2023) and Kim & Suzuki (2025), where CoT refers to autoregressive generation of intermediate tokens that decompose a complex computation into simpler steps, with teacher forcing during training. This is the same formal mechanism studied in complexity-theoretic analyses of CoT (Merrill & Sabharwal, 2024; Li et al., 2024), where "CoT" refers to the transformer generating intermediate computation steps, instead of natural language prompting.
>
> Nevertheless, we acknowledge the reviewer’s comment that our work’s connection to CoT prompting in the Wei, et al. (2022) sense (involving LLM training and natural language reasoning) is indirect. To address this concern, we will implement the following revisions:
>
> * We will clarify in the title and introduction that we study the \textit{theoretical mechanism} of CoT (autoregressive intermediate-step generation) instead of CoT prompting for LLMs.
> * We will add an explicit discussion section to distinguish our formal CoT definition from the natural language prompting definition, while noting that both settings share the identity as intermediate token generation.
> * We will restrain our claims to focus on the valid connection to the broader theoretical CoT literature instead of direct implications for LLMs.
>
> We would like to note that studying simplified models is standard methodology in the theory of deep learning. Previous works such as Zhang et al. (2024), Kim & Suzuki (2024, 2025), and Huang et al. (2024) all analyzed one-layer transformers to derive insights about attention mechanisms. Our work follows this pattern and extends it by introducing data shifts, a dimension no prior works have studied theoretically.
>
> > The relevance of its findings is limited to a narrow audience (likely only those prior works cited in the paper related to the k-parity problem and transformer architectures).
>
> We respectfully note that the theoretical CoT community has grown substantially in recent years, with a substantial number of papers at ICML, NeurIPS, and ICLR (Merrill & Sabharwal 2024; Li et al. 2024; Kim & Suzuki 2025; Wies et al. 2023). Our results speak to anyone interested in the robustness of CoT-style computation. The finding that supposedly helpful information leakage actually hurts CoT (Corollary 4.3) and the O(1/k) poisoning tolerance are both qualitative and quantitative insights relevant beyond the $k$-parity problem. We will add discussion connecting these findings to broader robustness concerns in multi-step reasoning.
>
> > [minor] Sec 3.1. Specification of the k-parity problem could be more precise, e.g., what is x_j, where it first appears? what exactly is the task in the current notation?
>
> We will add explicit definitions: $x_j ∈ {±1}$ denotes the $j$-th coordinate of the input vector $x \in \{ \pm 1 \}^d$, and the task is to predict $y = \prod_{j \in P} x_j$ given input $x$.

---

> ### Author Response · Authors · 2026-04-19
> **Responses to Reviewer REZ6 (Continued)**
>
> > [minor] Sec 3.2. There are several assumptions here, e.g., "we assume k=2^v for simplicity", are they without loss of generality? Please comment.
>
> This assumption is for analytical convenience and is standard in the literature (Wies et al., 2023; Kim & Suzuki, 2025). For general $k$, one can pad with dummy bits set to 1 to reach the next power of 2 (say, $k=9$, then we pad one in all remaining 7 bits until $2^4 = 16$) without changing the parity value. We will add a paragraph explaining this explicitly.
>
> > [minor] Sec. 4.2 is very difficult to follow, especially when it gets to "Examples for the cases.", the expressions for the gradients are non-obvious, but shown without any explanation.
>
> > [minor] Move experimental results into the body of the paper, for space, suggest moving more of the derivations in 4.2 to appendix.
>
> Thanks for pointing this out. We will add more intuitive explanations for each of the four gradient cases before presenting the formal expressions. We will also follow the reviewer's suggestion to move detailed derivations to the appendix and bring the experimental results (currently in Appendix C) into the main body.
>
> > [nit] Sec. 3 notation "any integer n" -> "any positive integer n > 0"?
>
> Thanks for pointing this out. We will modify "any integer $n$" to "any positive integer $n$."

---

### Decision · Action_Editor_k5Fc · 2026-06-25

**Recommendation:** Accept with minor revision

**Additional Comments:**

The decision is minor revision. The core required change is to match the scope of the paper's claims to the setting actually analyzes. The specific items to address:

1. Incorporate the additions promised in the response to Reviewer BfhY, in particular: (i) a more precise statement in the related work of how the contribution differs from Kim & Suzuki (2025); and (ii) an explicit account of how the proofs are specific to the architecture class analyzed.
2. Following Reviewers REZ6 and L7ve, limit the scope of the conclusions to the toy model of chain-of-thought established in the prior work of Wies et al. (2023) and Kim & Suzuki (2025). The extension to chain-of-thought in general may be presented as a possibility, but should be clearly flagged as not directly evidenced by the present work.
3. Reference related and recent empirical preprints on data poisoning: [Foerster et al. (2025)](https://arxiv.org/abs/2509.05739); [Chaudhari et al. (2026)](https://arxiv.org/abs/2601.19061), first appearing online in [Sept. 2025](https://openreview.net/forum?id=hSLopCTOtT).
4. Make the remaining minor clarifications raised across the reviews, including a comparison distinguishing label noise from data poisoning.

**Audience:**

Yes

**Audience Explanation:**

Reviewer BfhY notes: "Understanding the effect of the data distribution on post-training is an active area with great potential practical impact." I concur.

**Claims And Evidence:**

Yes

**Claims Explanation:**

All reviewers found the results in the context of the narrowed problem ($k$-(sparse )parity) and architecture class (one layer transformers with specific output construction) sound, as articulated in their reviews. While Reviewer L7ve submitted a final assessment that the claims in the paper do not meet its evidence, their comment conveys that this is about the general scope (chain-of-thought in large language models) targeted in the first submission version. Matching the scope of the claims to the actual setting considered is addressable in a minor revision, and the authors have already made progress on this in their author responses.